Resource

# Intestinal single-cell atlas reveals novel lymphocytes in pigs with similarities to human cells

Jayne E Wiarda[1,2,3] , Julian M Trachsel[1] , Sathesh K Sivasankaran[1,4], Christopher K Tuggle[5] , Crystal L Loving[1]

**Lymphocytes can heavily influence intestinal health, but resolving intestinal lymphocyte function is challenging as the intestine contains a vastly heterogeneous mixture of cells. Pigs are an advantageous biomedical model, but deeper understanding of intestinal lymphocytes is warranted to improve model utility. Twenty-six cell types were identified in the porcine ileum by single-cell RNA sequencing and further compared with cells in human and murine ileum. Though general consensus of cell subsets across species was revealed, some porcine-specific lymphocyte subsets were identified. Differential tissue dissection and in situ analyses conferred spatial context, revealing similar locations of lymphocyte subsets in Peyer's patches and epithelium in pig-to-human comparisons. Like humans, activated and effector lymphocytes were abundant in the ileum but not periphery of pigs, suggesting tissue-specific and/or activation-associated gene expression. Gene signatures for peripheral and ileal innate lymphoid cells newly discovered in pigs were defined and highlighted similarities to human innate lymphoid cells. Overall, we reveal novel lymphocyte subsets in pigs and highlight utility of pigs for intestinal research applications.**

## Introduction

The intestine is a selectively permeable barrier that absorbs nutrients while simultaneously limiting entry of potentially harmful external organisms and compounds. Thus, the intestinal immune system continuously deciphers between innocuous and dangerous stimuli. Coordination of immune responses is crucial for maintaining intestinal homeostasis; dysregulation of even a small number of cells can negatively impact intestinal health, as evidenced in nonpathogenic inflammatory conditions such as celiac disease, Crohn's disease, and ulcerative colitis (reviewed by Mowat and Agace [2014], Caio et al [2019], Caminero and Pinto-Sanchez [2020], and Caruso et al [2020]). Intestinal lymphocytes include B cells, T cells, and innate lymphoid cells (ILCs). The importance of lymphocytes in promoting intestinal homeostasis is well-documented in cases of intestinal dysfunction in individuals naturally lacking at least some lymphocyte populations (reviewed by Agarwal and Cunningham-Rundles [2019]) or experimental models where lymphocytes are depleted or immune pathways disrupted (Kühn et al, 1993; Mombaerts et al, 1993; Sadlack et al, 1993; Strober & Ehrhardt, 1993; Gärdby & Lycke, 2000; Laroux et al, 2004; Hepworth et al, 2015; Wang et al, 2017). Lymphocytes can be directed to provide protective adaptive immunity through mucosal vaccination strategies (reviewed by Li et al [2020] and Lavelle and Ward [2021]), whereas immune protection against a broad range of microorganisms may be achieved through nonconventional innate memory in some lymphocytes (reviewed by Wang et al [2019b]). Resultingly, there is pan-disciplinary interest in promoting health through modulation of intestinal lymphocytes, but decoding the complexity and function of these cells is an ongoing challenge. The intestine is a site of vast immune cellular diversity difficult to holistically characterize, yet defining heterogeneity within the cellular landscape of intestinal immune cells, such as lymphocytes, is one initial step to be taken toward better understanding intestinal immune dynamics and resulting effects on health.

Pigs (*Sus scrofa*) are a promising biomedical model and major global food source, yet the porcine intestinal immune cell landscape is poorly defined relative to humans and rodent models. Deeper exploration of the porcine intestinal immune system, particularly intestinal lymphocytes, will enhance utility of pigs as a well-defined and highly comparable biomedical model for gut health and/or disease (reviewed by Gonzalez et al [2015], Roura et al [2016], Ziegler et al [2016], and Käser [2021]) as pigs have greater physiologic and genetic similarities to humans than rodent models and are less expensive and more easily obtained than nonhuman primates (reviewed by Swindle et al [2011], Gün and Kues [2014], and Kobayashi et al [2018]). Enhanced characterization of porcine intestinal lymphocytes will also provide insight into promoting gut health and associated overall pig health to ultimately decrease disease susceptibility and strengthen pork as a major global food source. Though previous work has described porcine lymphocytes at the protein level (reviewed by Piriou-Guzylack and Salmon [2008]),

[1]Food Safety and Enteric Pathogens Research Unit, National Animal Disease Center, Agricultural Research Service, United States Department of Agriculture, Ames, IA, USA [2]Immunobiology Graduate Program, Iowa State University, Ames, IA, USA [3]Oak Ridge Institute for Science and Education, Agricultural Research Service Participation Program, Oak Ridge, TN, USA [4]Genome Informatics Facility, Iowa State University, Ames, IA, USA [5]Department of Animal Science, Iowa State University, Ames, IA, USA

Correspondence: crystal.loving@usda.gov

annotations are confined by a limited toolbox of available porcine protein-specific immunoreagents (reviewed by Entrican et al [2020]). Thus, definitions of porcine lymphocytes lack cellular resolution comparable to that of humans. This is particularly true for B cells, as a pan-B cell–specific extracellular protein marker is not available (reviewed by Piriou-Guzylack and Salmon [2008] and Sinkora and Butler [2009]), and ILCs, for which only natural killer (NK) cells have been identified (reviewed by Gerner et al [2009]). Approaches to resolve the porcine immune cell landscape at the transcriptional level have also been employed; however, traditional bulk RNA sequencing (RNA-seq) or microarray approaches fail to provide cellular resolution needed to decode such a complex cellular community (Herrera-Uribe et al, 2021), especially when immunoreagents for sorting of cells into more homogenous populations are lacking. Numerous studies have assessed transcriptional dynamics in the porcine intestinal tract but did not attempt to deconvolute cells into specific populations, a critical step in understanding functions of specific cells (Wang et al, 2008, 2019a; Freeman et al, 2012; Mach et al, 2014; Zhu et al, 2014; Inoue et al, 2015; Tan et al, 2017; Maroilley et al, 2018; Beiki et al, 2019; Meng et al, 2020; Summers et al, 2020; Jin et al, 2021; Pan et al, 2021). Some bulk RNA-seq studies have sorted porcine immune cells into specific populations based on cell surface markers but primarily focused on studying cells from the periphery and non-intestinal tissues (Auray et al, 2016, 2020; Foissac et al, 2019; Herrera-Uribe et al, 2021; Kim et al, 2021). Consequently, it remains to be determined whether existing data adequately portray the transcriptional heterogeneity of intestinal immune cells or if novelties exist in the context of the porcine intestine.

Single-cell RNA-seq (scRNA-seq) has been used to describe porcine immune cell transcriptomes at granularity unparalleled by bulk RNA-seq or microarray approaches, including in peripheral blood (Herrera-Uribe et al, 2021), lung (Zhang et al, 2021b), skin (Han et al, 2022), brain (Zhu et al, 2021), and embryos (Ramos-Ibeas et al, 2019; Kong et al, 2020; Liu et al, 2021). In addition, epithelial cells in the porcine intestine were recently queried via scRNA-seq, and results provide new insight into biological development and epithelial cell functions (Meng et al, 2021). However, high-resolution transcriptomic analysis of porcine intestinal immune cells remains to be completed. We therefore used scRNA-seq to provide the first high-resolution, global transcriptomic profiles of porcine intestinal lymphocytes. Interrogation was focused to the ileum, the most distal segment of the small intestine, which contains a unique combination of not only lymphocytes residing in the lamina propria and epithelium but also lymphocytes found in association with gut-associated lymphoid tissue (GALT) called Peyer's patches. Peyer's patches are major sites of immune induction not highly prevalent in other intestinal segments (Keren et al, 1978; Fujihashi et al, 2001; Mora et al, 2003; Kwa et al, 2006; Kiriya et al, 2007; Nagai et al, 2007; Bonnardel et al, 2015). In pigs, ileal Peyer's patches present as a continuous longitudinal strip along the length of the distal small intestine (Binns & Licence, 1985; Rothkötter, 2009) and are more easily identified and obtained compared with Peyer's patches in humans and rodents, the species in which scRNA-seq approaches have been mostly employed. Consequently, pigs are an ideal candidate for studying Peyer's patches because of easier gross identification and isolation for further study, but comparability of cells in porcine versus human Peyer's patches need be determined.

By performing scRNA-seq on porcine ileal-derived cells, we documented and showcased previously undescribed levels of cellular heterogeneity for multiple populations of lymphocytes and some non-lymphocytes. Profiling of porcine ileal cells was completed with multiple approaches, including cross-location and cross-species analyses. Data were compared with an annotated reference scRNA-seq dataset of porcine PBMCs (Herrera-Uribe et al, 2021) to reveal transcriptional differences between porcine intestinal-derived cells and circulating counterparts. Comparison to human and murine ileum reference datasets (Xu et al, 2019; Elmentaite et al, 2020) unveiled similarities and differences for cells of the same intestinal location across species. We further recognized cells associated specifically with Peyer's patches or the epithelium/lamina propria and confirmed findings by in situ and ex vivo detection using available canonical cell markers with locational context to further infer potential cell functions. Previously undescribed lymphocyte populations in pigs, particularly intestinal ILCs, were identified and characterized. We further leveraged our single-cell gene expression profiles to develop new cell marker combinations with currently available immunoreagents to label novel populations. ILC locational context within the ileum was determined, and transcriptional distinctions from circulating NK cells were denoted. Collectively, the data serve as a transcriptomic atlas of the porcine intestinal immune landscape resolved at the highest level of resolution (i.e., single-cell) to date and may be used to further decode cellular phenotype and function within the intestinal tract. To address research questions outside of the scope of this work, data are also available for interactive, online query (see Data Availability section).

# Results

### Experimental overview

From each of two pigs, the distal ileum was grossly dissected into three distinct sections for cell isolation: (1) ileal tissue containing only regions with Peyer's patches (PP), (2) ileal tissue excluding regions with Peyer's patches (non-PP), and (3) a complete cross section of ileal tissue containing both regions (whole ileum; Figs 1A and S1A and B). For each region, a single-cell suspension of combined epithelium, lamina propria, Peyer's patches (if present), and submucosa was retrieved, enriched for viable lymphocytes, and submitted for scRNA-seq, as described in the Materials and Methods section. Sequencing and further processing/quality control of scRNA-seq data are fully described in the Materials and Methods section and are shown in Fig S2A–E and Table S1. Our final dataset contained 31,983 total cells from six ileal samples (Fig 1B). Cells were classified into four cell lineages and further annotated as 26 cell types (Figs 1C and D and S3) using a multi-method annotation approach described fully in the Materials and Methods section and shown in Figs S4–S10 and Tables S2–S7. Cell type annotations were based on biological interpretation of genes encoding for both phenotypic and functional markers.

Annotated porcine ileal cells were next treated as query data for comparison to existing scRNA-seq datasets (as described in the

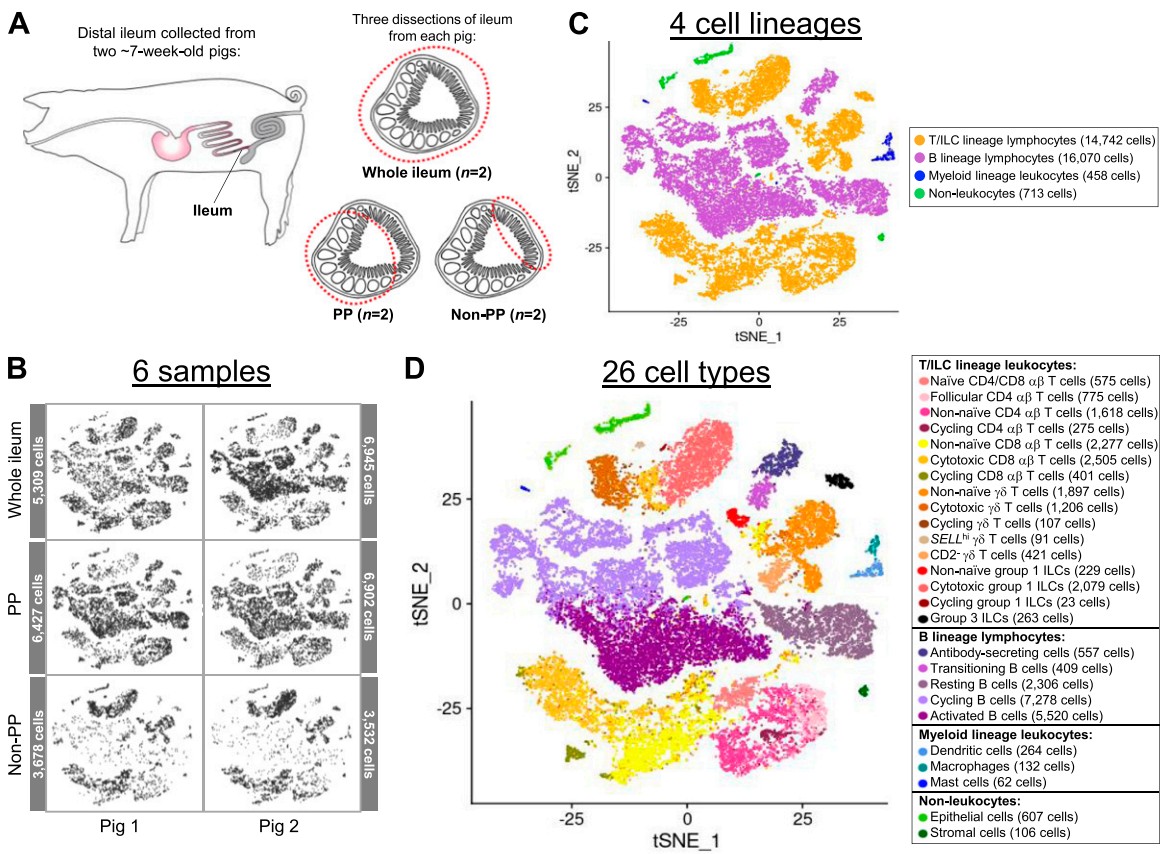

**Figure 1.  Experimental overview and annotation of cells recovered from scRNA-seq of the porcine ileum.**
**(A)** Ileal samples collected from two 7-wk-old pigs for scRNA-seq. Left: representative image of tissue collection site from the ileum of the distal small intestine within the porcine gastrointestinal tract. Right: representative images of tissue dissections from transverse cross sections of the ileum. Dissections from each pig included a cross section of the whole ileum including areas with and without Peyer's patches (whole ileum), the ileum containing only regions with Peyer's patches (PP), and the ileum containing only regions without Peyer's patches (non-PP), resulting in a total of six samples processed for scRNA-seq. **(A)** Two-dimensional t-SNE visualization of 31,983 cells isolated from porcine ileal samples described in (A), subjected to scRNA-seq, and included in the final dataset following data processing and quality filtering. Each point represents a single cell. **(B, C, D)** Plots show which sample cells are derived from (B) and cell lineage (C) or cell type (D) annotations. In (B), cells in individual panels are derived from a specified sample. In (C, D), the color of a cell indicates cell lineage (C) or cell type (D) annotation. **(B, C, D)** The number of cells belonging to each sample (B), cell lineage (C), and cell type (D) are listed next to corresponding panels. Abbreviations: ILC, innate lymphoid cell; PP, Peyer's patch; scRNA-seq, single-cell RNA sequencing; tSNE, t-distributed stochastic neighbor embedding.

Materials and Methods section, "*Reference-based label transfer and mapping*") to provide greater insight into annotated cell identities. Because a comparable porcine intestinal scRNA-seq dataset was not available, scRNA-seq reference data of healthy porcine PBMCs (Herrera-Uribe et al, 2021), human ileum (Elmentaite et al, 2020), and murine ileum (Xu et al, 2019) were used to provide intraspecies/inter-tissue and interspecies/intra-tissue comparisons. Degree of similarity between query and reference cells was determined by calculating mapping scores via reference-based cell mapping (Fig S11). Transfer of cell labels from reference onto query single cells provided prediction probabilities to cell types described in each reference dataset (Fig S12). Gene expression profiles, enrichment of biological processes, and reference-based mapping and cell type prediction results for lymphocytes are presented in the next two results sections.

The main purpose of this work was to deeply characterize porcine ileal lymphocytes, and most of the cells across all six ileal samples were annotated as belonging to B (50.25%) or T/ILC (46.09%) lymphocyte lineages. However, some non-lymphocytes were also identified, including myeloid lineage leukocytes (1.43%) and non-leukocytes (2.23%). Myeloid lineage leukocytes were composed of DCs (264 cells), macrophages (132 cells), and mast cells (62 cells). Identified non-leukocytes included epithelial (607 cells) and stromal (106 cells) cells. Because characterization of non-lymphocytes was not our primary intent for this work, non-leukocytes are not discussed further, but data are available for deeper inquiry (see Fig S10 and Tables S6 and S7 and our Data Availability section).

### Defining the porcine ileal immune landscape: T cells and ILCs

Similar to scRNA-seq results described elsewhere (Zhao et al, 2020; Elmentaite et al, 2021; Guo et al, 2021; Herrera-Uribe et al, 2021; Patel et al, 2021), T cells and ILCs were so transcriptionally similar to one another that they were annotated into a single cell lineage and further resolved into 16 cell types (Figs 2A and S13). T cells were identified by expression of the porcine pan-T cell marker *CD3E*

(reviewed by Piriou-Guzylack and Salmon [2008] and Gerner et al [2009]) and included subsets of CD4 $\alpha\beta$, CD8 $\alpha\beta$, and $\gamma\delta$ T cells expressing *CD4*, *CD8B*, and *TRDC*, respectively. ILCs largely lacked *CD3E* and included subsets of group 1 and group 3 ILCs based on expression of genes associated with type 1 or type 3 immunity, respectively (described in subsequent cell type descriptions below; Fig 2B). By hierarchical analysis, T/ILC types were more closely related by inferred function (e.g., cell cycling, activation, and cytotoxicity) rather than traditional T/ILC phenotypes (e.g., CD4 $\alpha\beta$ T cells, CD8 $\alpha\beta$ T cells, $\gamma\delta$ T cells, group 1 ILCs, group 3 ILCs; Fig 2B), which are classically defined based on expression of a series of cell surface markers.

### Cycling T cells and ILCs

One hierarchical grouping of T/ILC types in Fig 2B was composed of all cycling T/ILCs, including cycling CD4 $\alpha\beta$ T, CD8 $\alpha\beta$ T, $\gamma\delta$ T, and group 1 ILCs. All cycling cells had significantly increased expression of genes associated with replication/division (e.g., *PCLAF*, *BIRC5*, *TOP2A*, *STMN1*; Dabydeen et al, 2019; Giotti et al, 2019) and enrichment of related biological processes (e.g., establishment of mitotic spindle orientation [*GO:0000132*], regulation of mitotic centrosome separation [*GO:0046602*], DNA duplex unwinding [*GO:0032508*]) relative to other T/ILC types (Fig 2B and Table S8). Cycling T/ILCs had highest average mapping scores to reference porcine PBMCs (≥0.737), followed by human ileum (≥0.699) and murine ileum (≥0.664; Figs 2C and S14). Though predictions of many cycling T/ILC types were to similarly annotated T/ILCs in reference datasets (e.g., porcine ileal cycling CD4 $\alpha\beta$ T cells having highest average predictions to reference CD4 $\alpha\beta$ T cell types; Figs 2D and S15–S17), several cycling T/ILC types had high prediction to B cells in porcine PBMCs or cycling B cells in the human ileum (Figs S15 and S16). For instance, cycling CD8 $\alpha\beta$ T, $\gamma\delta$ T, and group 1 ILCs all had first or second highest average prediction probabilities to cycling B cells in human ileum (Fig S16), indicating cycling T/ILCs share transcriptional similarities to cycling B cells in the human ileum, likely because of shared replication/division-specific gene expression as opposed to shared expression of genes involved in lymphocyte lineage-specific immune functions of the cell.

### Cytotoxic T cells and ILCs

Cytotoxic CD8 $\alpha\beta$ T, $\gamma\delta$ T, and group 1 ILCs were most closely related to one another and had significantly elevated expression of genes encoding for cytotoxic molecules, including *GZMA*\* (\*Ensembl identifiers found in gene annotation were converted to gene symbols; refer to the Materials and Methods section "*Gene name modifications*" for more details), *GZMB*, and *GNLY* (Hidalgo et al, 2008), relative to other T/ILC types (Fig 2B and Table S8). The biological process leukocyte mediated cytotoxicity (*GO:0001909*) was enriched in cytotoxic CD8 $\alpha\beta$ T cells, whereas regulation of natural killer cell–mediated cytotoxicity (*GO:0042269*) was enriched in cytotoxic $\gamma\delta$ T and group 1 ILCs (Table S8). Cytotoxic cell types had some of the lowest average mapping scores to reference porcine PBMCs (range of means 0.645–0.732), indicating dissimilarity between cytotoxic ileal cells from any cells in circulation (Figs 2C and S14). Though cytotoxic cell types had lower mapping scores to porcine PBMCs, cytotoxic CD8 $\alpha\beta$ T cells and group 1 ILCs still had the highest prediction to comparable cell types in porcine

peripheral blood: CD8 $\alpha\beta^+$ $\alpha\beta$ T cells and NK cells, respectively (Figs 2D and S15). Ileal cytotoxic $\gamma\delta$ T cells had highest average prediction to innate-like CD8$\alpha^+$ $\alpha\beta$ T cells and NK cells from porcine peripheral blood rather than to peripheral CD2$^+$ $\gamma\delta$ T cells, further supporting poor representation of ileal cytotoxic $\gamma\delta$ T cells by porcine peripheral $\gamma\delta$ T cells and suggesting greater similarities to other peripheral innate or innate-like T/ILC types instead (Figs 2D and S15). A similar pattern was observed in murine ileum, where porcine ileal cytotoxic $\gamma\delta$ T cells had the highest prediction to reference NK cells rather than *Gzma+* $\gamma\delta$ T cells, suggesting again that porcine ileal cytotoxic $\gamma\delta$ T cells had greater similarity to other innate/innate-like T/ILC types rather than $\gamma\delta$ T cells (Figs 2D and S17).

### Non-naive $\gamma\delta$ T cells, CD8 $\alpha\beta$ T cells, and group 1 ILCs

Non-naive $\gamma\delta$ T, CD8 $\alpha\beta$ T, and group 1 ILCs also formed a hierarchical grouping closely related to cytotoxic cell counterparts in Fig 2B. In contrast to cytotoxic T/ILCs, non-naive $\gamma\delta$ T, CD8 $\alpha\beta$ T, and group 1 ILCs had lower expression of genes encoding cytotoxic molecules (e.g., *GZMA\**, *GZMB*, and *GNLY*) but significantly elevated expression of other genes indicative of previous or recent cell activation, including *CTSW*, *XCL1*, *SLA-DRA\**, *SLA-DQB1*, and *CCR9* (Fig 2B and Table S8; Kelner et al, 1994; Boismenu et al, 1996; Svensson et al, 2002; Uehara et al, 2002; Iwata et al, 2004; Ondr & Pham, 2004; Gerner et al, 2009; Stoeckle et al, 2009). Non-naive $\gamma\delta$ T, CD8 $\alpha\beta$ T, and group 1 ILCs were all enriched for the biological processes positive regulation of T cell differentiation (*GO:0045582*) and positive regulation of T cell–mediated immunity (*GO:0002711*), further supporting a non-naive cell state (Table S8). Non-naive $\gamma\delta$ T, CD8 $\alpha\beta$ T, and group 1 ILCs had higher average mapping scores to all reference datasets (range of means 0.872–0.902) than did corresponding cytotoxic T/ILCs, indicating better representation by reference data (Figs 2C and S14). Unlike cytotoxic $\gamma\delta$ T cells, non-naive $\gamma\delta$ T cells had the highest prediction to $\gamma\delta$ T cell types in reference datasets, including CD2$^+$ $\gamma\delta$ T cells (porcine PBMCs), $\gamma\delta$ T/NK cells (human ileum), and *Xcl1+* $\gamma\delta$ T cells (murine ileum), suggesting greater similarity of porcine ileal non-naive $\gamma\delta$ T cells to reference $\gamma\delta$ T cell populations than observed for porcine ileal cytotoxic $\gamma\delta$ T cells (Figs 2D and S15–S17). Non-naive CD8 $\alpha\beta$ T cells had highest average predictions to CD8 $\alpha\beta$ T cell types in reference datasets, as did non-naive group 1 ILCs to reference ILC types (Figs 2D and S15–S17). Though both non-naive and cytotoxic group 1 ILCs had the highest prediction to reference group 1 ILC types, activated group 1 ILCs had the highest prediction to porcine peripheral CD8$\alpha^+$ $\alpha\beta$ T/NK cells and murine ileal ILC1s. In contrast, cytotoxic group 1 ILCs had the highest prediction to NK cells in the same reference datasets, delineating transcriptional distinctions between cytotoxic and non-naive group 1 ILCs that correspond better to different reference cell types (Figs 2D, S15, and S17).

### SELL$^{hi}$ $\gamma\delta$ T cells

SELL$^{hi}$ $\gamma\delta$ T cells were a minor fraction of porcine ileal $\gamma\delta$ T cells (91 cells total) that shared a node with cytotoxic and non-naive T/ILCs expressing effector/activation molecules, including *CCL5* and *ITGAE* (Fig 2B; Ling et al, 2007; Szabo et al, 2019a). SELL$^{hi}$ $\gamma\delta$ T cells nearly ubiquitously expressed *SELL* (encoding CD62L) and genes related to cytotoxicity (e.g., *GZMA\** and *GZMB*) but also some genes expressed by non-naive T/ILCs, such as *XCL1* (Fig 2B). SELL$^{hi}$ $\gamma\delta$ T cells

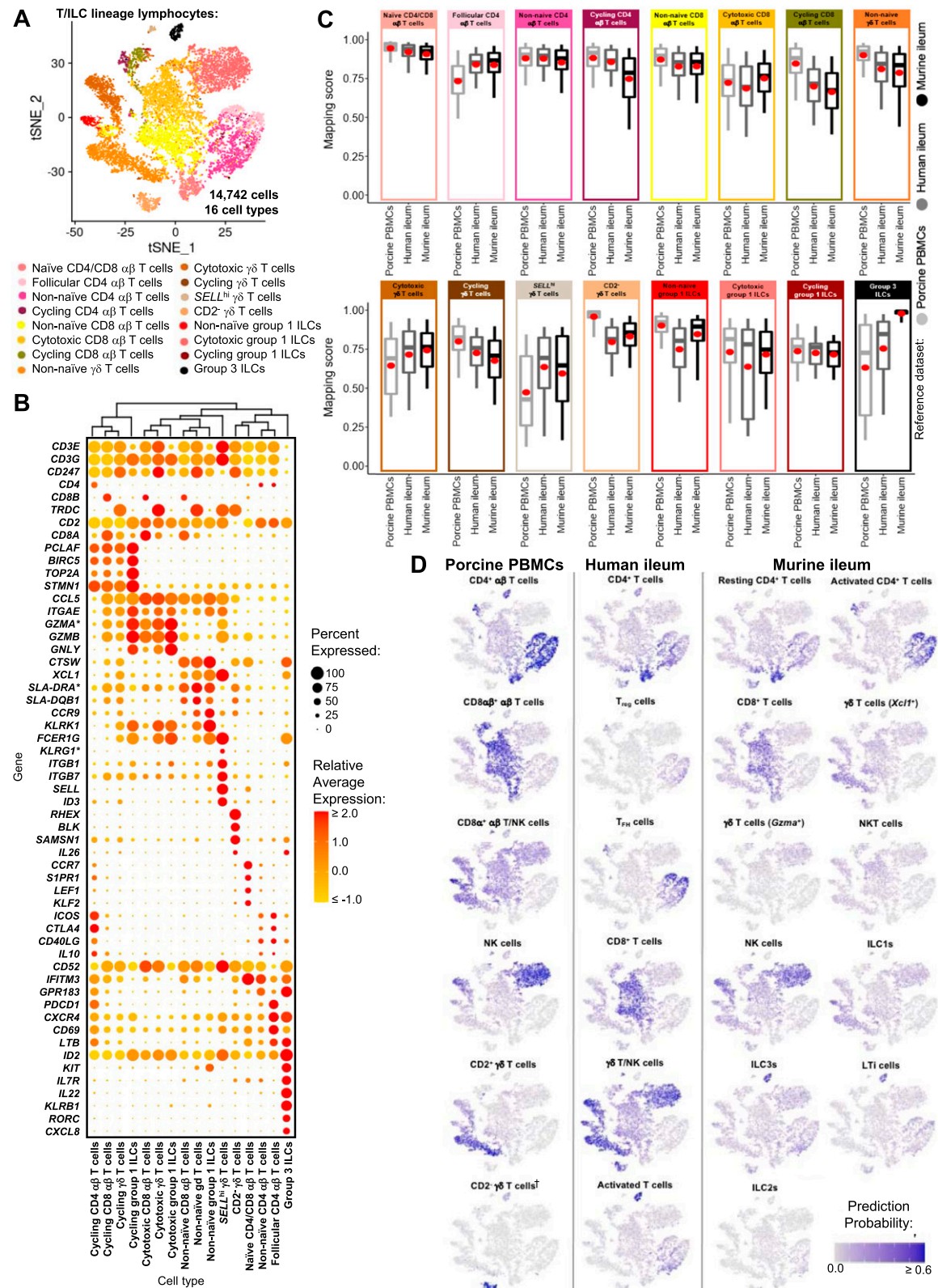

**Figure 2. scRNA-seq profiles of T/ILC lineage lymphocytes in the porcine ileum.**
**(A)** Two-dimensional t-SNE visualization of 14,742 cells recovered from the porcine ileum via scRNA-seq that were classified as T/ILC lineage lymphocytes in Figs 1C and S4B and further annotated into 16 cell types in Figs 1D and S5–S8. Each point represents a single cell; the color of each point indicates cell types shown in Fig 1D. **(B)** Hierarchical relationship of T/ILC lineage lymphocyte cell types from the porcine ileum shown in a dendrogram (upper) and expression patterns of selected genes within

expressed genes encoding innate receptors, including *FCER1G* and *KLRG1\**, but lacked expression of others, such as *KLRK1* (Fig 2B). Moreover, *SELL*hi *γδ* T cells had significantly higher expression of genes encoding for adhesion molecules (e.g., *SELL*, *ITGB1*, and *ITGB7*) and the transcriptional regulator and *γδ* T cell fate determinator, *ID3* (Fig 2B and Table S8; Lauritsen et al, 2009). Five of the top eight enriched biological processes for *SELL*hi *γδ* T cells (as determined by smallest *P*-values) included actin filament depolymerization (*GO:0030042*), positive regulation of actin filament polymerization (*GO:003038*), establishment or maintenance of cell polarity (*GO:0007163*), integrin-mediated signaling pathway (*GO:0007229*), and natural killer cell activation (*GO:0030101*), indicating a highly activated state potentially related to cell receptor engagement/signaling (Table S8). *SELL*hi *γδ* T cells had the lowest average mapping scores of all cell types in comparison to each reference dataset (range of means 0.473–0.636; Figs 2C and S14), suggesting they were unique to the porcine ileum.

### CD2⁻ *γδ* T cells

CD2⁻ *γδ* T cells (characterized as *TRDC*-expressing cells that lacked *CD2* expression; Fig 2B) are present in pigs but absent from humans and mice (Stepanova & Sinkora, 2013). Correspondingly, CD2⁻ *γδ* T cells had higher average mapping scores to porcine PBMCs (0.959) than to ileal cells from human (0.796) or mouse (0.832; Figs 2C and S14). Several lines of work support CD2⁻ *γδ* T cells as a cell lineage separate from CD2⁺ *γδ* T cells (Sinkora et al, 2005, 2007; Stepanova & Sinkora, 2013; Sedlak et al, 2014; Rodríguez-Gómez et al, 2019; Hammer et al, 2020), whereas others have suggested CD2⁻ *γδ* T cells are naive cells in pigs (Stepanova & Sinkora, 2012; Talker et al, 2013; Käser, 2021). We found that CD2⁻ *γδ* T cells were distantly related from all other annotated *γδ* T cells (all considered CD2⁺ *γδ* T cells; Fig 2B), which could suggest that CD2⁻ *γδ* T cells are a distinct cell lineage from CD2⁺ *γδ* T cells. Contrarily, CD2⁻ *γδ* T cells were most closely related to naive CD4/CD8 *αβ* T cells in the porcine ileum by hierarchical clustering (Fig 2B), which could suggest CD2⁻ *γδ* T cells are naive cells. Regardless of whether CD2⁻ *γδ* T cells represent a distinct cell lineage or naive cells, CD2⁻ *γδ* T cells had the highest average mapping scores to porcine PBMCs of all T/ILC types, indicating CD2⁻ *γδ* T cells to be the porcine ileal T/ILC type most similar to cells in the porcine periphery. Besides lacking *CD2* expression, ileal CD2⁻ *γδ* T cells had significantly elevated expression of *RHEX*, *BLK*, *SAMSN1*, and *IL26* (Fig 2B and Table S8), which were also highly expressed by CD2⁻ *γδ* T

cells in the porcine periphery (Herrera-Uribe et al, 2021). CD2⁻ *γδ* T cells were predicted most similar to corresponding CD2⁻ *γδ* T cells in porcine peripheral blood and to *γδ* T/NK cells in human ileum, whereas in murine ileum, predictions were lowly distributed across multiple T/ILC subsets (Figs 2D and S15–S17). Thus, CD2⁻ *γδ* T cells can be found in both the ileum and periphery of pigs but do not have close counterparts in the human or murine ileum.

### Naive CD4/CD8 *αβ* T cells

Naive CD4 and CD8 *αβ* T cells had significantly higher expression of genes related to cell circulation and a naive T cell phenotype, including *CCR7*, *S1PR1*, *LEF1*, and *KLF2* (Fig 2B and Table S8; Willinger et al, 2006; Sebzda et al, 2008; Skon et al, 2013; Cano-Gamez et al, 2020; Shan et al, 2021). Of all T/ILCs, naive CD4/CD8 *αβ* T cells had the second-highest average mapping scores to porcine PBMCs (0.944; Figs 2C and S14), indicating naive CD4 and CD8 *αβ* T cells to be the porcine ileal T/ILC type second-best represented by cells in the porcine periphery, trailing only behind CD2⁻ *γδ* T cells. High mapping scores to human and murine ileum (means 0.921 and 0.917, respectively) were also noted, indicating good representation of naive CD4 and CD8 *αβ* T cells in the ileum of both human and mouse. Porcine ileal naive CD4/CD8 *αβ* T cells had the highest prediction to corresponding populations in reference datasets, including CD4 and CD8 *αβ* T cell populations derived from porcine PBMCs or human ileum and resting CD4 and CD8 T cells derived from murine ileum (Figs 2D and S15–S17).

### Non-naive and follicular CD4 *αβ* T cells

Remaining non-naive/non-cycling CD4 *αβ* T cells in the porcine ileum included follicular and non-naive CD4 *αβ* T cells, which were most closely related to one another (Fig 2B). Non-naive CD4 *αβ* T cells did not share elevated expression of several genes highly expressed by other non-naive T/ILC types (e.g., *CCL5*, *ITGAE*, *CTSW*, *XCL1*, *SLA-DRA\**, *SLA-DQB1*, and *CCR9*) but instead had significantly elevated expression of genes associated with CD4 *αβ* T cell activation (e.g., *ICOS*, *CTLA4*, and *CD40LG*; Jaiswal et al, 1996; Linsley & Golstein, 1996; Hutloff et al, 1999; Miragaia et al, 2019; Cano-Gamez et al, 2020), which were also elevated in follicular CD4 *αβ* T cells (Fig 2B and Table S8). However, non-naive CD4 *αβ* T cells had higher expression of activation-associated genes *IFITM3* and *GPR183* (Clottu et al, 2017; Bedford et al, 2019; Szabo et al, 2019a) relative to follicular CD4 *αβ* T cells. Follicular CD4 *αβ* T cells were characterized

each cell type shown in a dot plot (lower). In the dot plot, selected genes are listed on the y-axis, and cell types are listed on the x-axis. Within the dot plot, size of a dot corresponds to the percentage of cells expressing a gene within an annotated cell type; color of a dot corresponds to average expression level of a gene for those cells expressing it within a cell type relative to all other cells in the dataset shown in (A). **(C)** Box plots of the distribution of mapping scores for T/ILC lineage lymphocyte cell types from the porcine ileum mapped to each reference scRNA-seq dataset. Results for a single cell type are located within a single box, with color of the box corresponding to colors used for cell types in (A). The color of each box in a plot corresponds to the reference dataset porcine ileal cells were mapped to, including porcine PBMCs (light gray), human ileum (medium gray), and murine ileum (black). Boxes span the interquartile range (IQR) of the data (25th and 75th percentiles), with the median (50th percentile) indicated by a horizontal line. Whiskers span the 5th and 95th percentiles of the data. A red dot represents the data mean. **(D)** Prediction probabilities for porcine-ileum scRNA-seq query data from label transfer of selected annotated T/ILC types in reference scRNA-seq datasets of porcine PBMCs (left), human ileum (middle), and murine ileum (right) overlaid onto two-dimensional t-SNE visualization shown in (A). Each point represents a single cell; the color of each point indicates prediction probability to a corresponding cell type from reference data, as indicated directly above each t-SNE plot. A higher prediction probability indicates higher similarity to a specified annotated cell type in a reference scRNA-seq dataset. scRNA-seq data shown in (A, B, C, D) were derived from the ileum of two 7-wk-old pigs. *Ensembl identifiers found in gene annotation were converted to gene symbols; refer to methods section "*Gene name modifications*" for more details. † Identical cell type annotations were given to cells in both the porcine ileum and a reference scRNA-seq dataset. Cell type annotations were given to each dataset by independent rationales, and identical annotations do not necessarily indicate identical cell types were recovered from both porcine-ileum and reference data. Abbreviations: ILC, innate lymphoid cell; IQR, interquartile range; LTi, lymphoid tissue inducer; NK, natural killer; NKT, natural killer T; PBMC, peripheral blood mononuclear cell; scRNA-seq, single-cell RNA sequencing; t-SNE, t-distributed stochastic neighbor embedding; T_FH, T follicular helper; T_reg, T regulatory.

by higher expression of *PDCD1*, *CXCR4*, *CD69*, and *LTB*, all genes highly expressed by follicle-associated T cells (Schaerli et al, 2000; Haynes et al, 2007; Shi et al, 2018), such as T follicular helper ($T_{FH}$) or T follicular regulatory ($T_{FR}$) cells (Fig 2B). The top two enriched biological processes in follicular CD4 $\alpha\beta$ T cells (smallest *P*-values) were related to B cell activation/humoral immunity, including humoral immune response mediated by circulating immunoglobulin (*GO:0002455*) and plasma cell differentiation (*GO:0002317*; Table S8). Follicular CD4 $\alpha\beta$ T cells had lower mapping scores to porcine PBMCs (mean 0.733) than did non-naive CD4 $\alpha\beta$ T cells (mean 0.880; Figs 2C and S15), indicating greater dissimilarity of follicular CD4 $\alpha\beta$ T cells than non-naive CD4 $\alpha\beta$ T cells to circulating cells in pigs. Porcine follicular CD4 $\alpha\beta$ T cells had the highest prediction to $T_{FH}$ cells in human ileum and activated CD4 T cells in murine ileum (Figs 2D, S16, and S17), further supporting an activated role associated with follicular helper/regulatory functions. Non-naive CD4 $\alpha\beta$ T cells were largely predicted as activated CD4 T cells in murine ileum and more so as CD4 T than $T_{FH}$ in human ileum (Figs 2D, S16, and S17), supporting an activated cell state.

### Group 3 ILCs

Group 3 ILCs expressed many genes characteristic of type 3 immunity, including *IL22*, *RORC*, and *CXCL8* (Schaerli et al, 2000; Haynes et al, 2007; Shi et al, 2018; Qi et al, 2021) and were more closely related to non-cycling CD4 $\alpha\beta$ and naive T cell subsets than to any type of group 1 ILC (Fig 2B). Though ILCs largely lacked expression of pan-T cell marker *CD3E*, group 1 ILCs still expressed other CD3 complex-associated genes, such as *CD3G* and *CD247* (encoding CD3γ and CD3ζ, respectively). In contrast, group 3 ILCs largely lacked expression of all aforementioned CD3 subunit-encoding genes and also had significantly higher expression of classical ILC gene markers, including *KIT*, *ID2*, *IL7R*, and *KLRB1* (Fig 2B and Table S8; Yokota et al, 1999; Yoshida et al, 1999; Boos et al, 2007; Satoh-Takayama et al, 2010; Spits et al, 2013), though these markers are already known to be variably expressed by intestinal group 1 ILCs based on species and regional location (Robinette et al, 2015; Simoni et al, 2017; Simoni & Newell, 2017; Van Acker et al, 2017; Meininger et al, 2020). Group 3 ILCs mapped best to cells of murine ileum (mean mapping score 0.979) and were predicted most similar to corresponding group 3 ILC populations of ILC3s or lymphoid tissue inducer (LTi) cells (Figs 2C and D and S17). In contrast, group 3 ILCs did not have as close a counterpart in porcine PBMCs or human ileum, as indicated by lower average mapping scores (0.632 and 0.754, respectively) and prediction most similar to CD4 $\alpha\beta$ T cells or activated T cells, respectively (Figs 2C and D, S15, and S16).

### Defining the porcine ileal immune landscape: B cells and antibody-secreting cells (ASCs)

B lineage lymphocytes were annotated as ASCs, B cells transitioning into ASCs (referred to as transitioning B cells), and three additional populations of B cells, including resting, cycling, and activated B cells (Figs 3A and S9F).

### ASCs

ASCs were most distantly related from other B cell types by hierarchical clustering and had significantly lower expression of several canonical B cell genes, including *CD19*, *CD79A*, *CD79B*, *MS4A1*, and *PAX5* (Fig 3B and Table S9; Herrera-Uribe et al, 2021; Lee et al, 2021). Genes known to be expressed by porcine peripheral ASCs (e.g., *JCHAIN*, *XBP1*, *IRF4*, and *PRDM1*; Herrera-Uribe et al, 2021) had elevated expression in ileal ASCs as well. The top two enriched biological processes in ASCs relative to other B cells were related to B cell activation (positive regulation of B cell activation [*GO:0050871*]) and protein production, such as required for producing antibodies (positive regulation of protein exit from the endoplasmic reticulum [*GO:0070863*]; Table S9). ASCs were well-represented by all reference datasets, as indicated by high mapping scores (means ≥ 0.927; Figs 3C and S18) and were almost unanimously predicted as ASC/plasma cell types from all reference datasets (Figs 3D and S19–S21).

### Transitioning B cells

Similar to ASCs, transitioning B cells had high expression of genes characteristic of porcine ASCs, including *JCHAIN*, *XBP1*, *IRF4*, and *PRDM1*, and were enriched for biological processes supporting antibody production, including the top three enriched processes of posttranslational protein targeting to endoplasmic reticulum membrane (*GO:0006620*), protein N-linked glycosylation (*GO:0006487*), and glycoprotein catabolic process (*GO:0006516*; Fig 3B and Table S9). Porcine ileal ASCs had highest average prediction to ASC/plasma cell types in reference datasets; however, prediction scores to ASC/plasma cell types for transitioning B cells were lower than those observed in ASCs, and transitioning B cells also had high prediction to activated B cells in human ileum (Figs 3D and S19–S21). Transitioning B cells had lower average mapping scores to all reference datasets than did ASCs (means ≥ 0.718; Figs 3C and S18), indicating poorer representation by the reference data. In contrast to ASCs, transitioning B cells had greater expression of canonical B cell genes (e.g., *CD19*, *CD79A*, *CD79B*, *MS4A1*, and *PAX5*) and higher expression of several markers of early B cell activation, including *CD69*, *CD83*, *SLA-DQB1*, and *SLA-DRA** (Fig 3B; Van der Stede et al, 2005; Breloer et al, 2007; Prazma et al, 2007; Ashouri & Weiss, 2017; Rahe & Murtaugh, 2017), supporting functional inference that cells were a subset of more recently activated B cells transitioning to produce and secrete antibody.

### Resting B cells

Remaining B cell types (resting, cycling, and activated B cells) all had greater expression of B cell canonical genes (e.g., *CD19*, *CD79A*, *CD79B*, *MS4A1*, and *PAX5*) than did ASCs and lesser expression of aforementioned genes expressed by both ASCs and transitioning B cells (e.g., *JCHAIN*, *XBP1*, *IRF4*, and *PRDM1*; Fig 3B). Resting B cells were most closely related to transitioning B cells in porcine ileum but, unlike remaining cycling/activated B cell subsets, lacked expression of several genes associated with activation and/or germinal centers, including *AICDA*, *BCL6*, and *CD86* (Engel et al, 1994; Allman et al, 1996; Muramatsu et al, 1999; Lee et al, 2021), indicating cells in a resting state (Fig 3B). Resting B cells had increased expression of genes characteristic of cell circulation and naive/memory B cells, including *KLF2*, *SELL* (CD62L), *CCR7*, *FCER2* (CD23), and *CD40* (Fig 3B; Waldschmidt et al, 1988; Förster et al, 1999; Bhattacharya et al, 2007; Winkelmann et al, 2011; Rahe & Murtaugh,

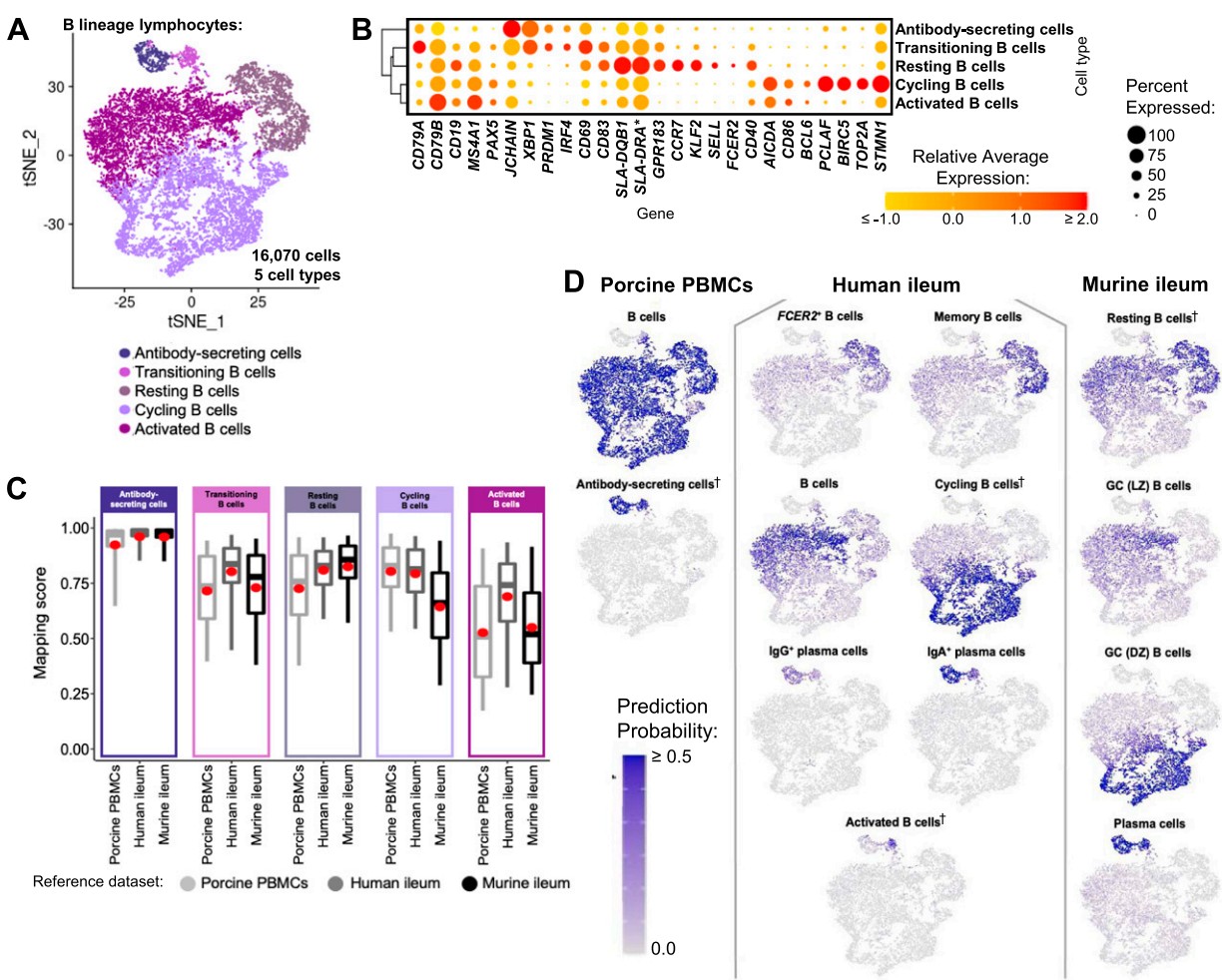

**Figure 3. scRNA-seq profiles of B lineage lymphocytes in the porcine ileum.**
**(A)** Two-dimensional t-SNE visualization of 16,070 cells recovered from the porcine ileum via scRNA-seq that were classified as B lineage lymphocytes in Figs 1C and S4B and further annotated into five cell types in Figs 1D and S9. Each point represents a single cell; the color of each point indicates cell types shown in Fig 1D. **(B)** Hierarchical relationship of B lineage lymphocyte cell types from the porcine ileum shown in a dendrogram (left), and expression patterns of selected genes within each cell type shown in a dot plot (right). In the dot plot, selected genes are listed on the x-axis, and cell types are listed on the y-axis. Within the dot plot, size of a dot corresponds to the percentage of cells expressing a gene within an annotated cell type; color of a dot corresponds to average expression level of a gene for those cells expressing it within a cell type relative to all other cells in the dataset shown in (A). **(C)** Box plots of the distribution of mapping scores for B lineage lymphocyte cell types from the porcine ileum mapped to each reference scRNA-seq dataset. Results for a single cell type are located within a single box, with color of the box corresponding to colors used for cell types in (A). The color of each box in a plot corresponds to the reference dataset porcine ileal cells were mapped to, including porcine PBMCs (light gray), human ileum (medium gray), and murine ileum (black). Boxes span the interquartile range (IQR) of the data (25th and 75th percentiles), with the median (50th percentile) indicated by a horizontal line. Whiskers span the 5th and 95th percentiles of the data. A red dot represents the data mean. **(D)** Prediction probabilities for porcine-ileum scRNA-seq query data from label transfer of selected annotated B/antibody-secreting cell types in reference scRNA-seq datasets of porcine PBMCs (left), human ileum (middle), and murine ileum (right) overlaid onto two-dimensional t-SNE visualization shown in (A). Each point represents a single cell; the color of each point indicates prediction probability to a corresponding cell type from reference data, as indicated directly above each t-SNE plot. A higher prediction probability indicates higher similarity to a specified annotated cell type in a reference scRNA-seq dataset. scRNA-seq data shown in (A, B, C, D) were derived from ileum of two 7-wk-old pigs. *Ensembl identifiers found in gene annotation were converted to gene symbols; refer to methods section "*Gene name modifications*" for more details. † Identical cell type annotations were given to cells in both porcine ileum and a reference scRNA-seq dataset. Cell type annotations were given to each dataset by independent rationales, and identical annotations do not necessarily indicate identical cell types were recovered from both porcine-ileum and reference data. Abbreviations: DZ, dark zone; GC, germinal center; IQR, interquartile range; LZ, light zone; PBMC, peripheral blood mononuclear cell; scRNA-seq, single-cell RNA sequencing; t-SNE, t-distributed stochastic neighbor embedding.

2017; Zhang et al, 2021a; Lee et al, 2021); however, it remained indiscriminate as to whether resting B cells were naive, memory, or a combination of both as many of the same genes are expressed by both naive and memory B cell subsets. In comparison to reference datasets, resting B cells were mostly predicted as memory B cell types (memory or *FCER2*⁺ B cells) in human ileum and as resting B cells in murine ileum (Figs 3D, S20, and S21).

### Activated and cycling B cells
The remaining two B cell types in porcine ileum included cycling and activated B cells, which were most closely related to one another in Fig 3B. Both cell types had high expression of genes related to B cell activation and/or germinal center–associated responses (e.g., *AICDA*, *BCL6*, and *CD86*; Ye et al, 1997; Muramatsu et al, 1999; Victora et al, 2010), but cycling B cells also had

characteristics of cellular replication/division, including elevated expression of *PCLAF*, *BIRC5*, *TOP2A*, and *STMN1* (Dabydeen et al, 2019; Giotti et al, 2019) and enrichment of biological processes such as nucleosome organization (*GO:0034728*), centriole–centriole cohesion (*GO:0010457*), and mitotic spindle organization (*GO:0007052*; Fig 3B and Table S9). Porcine ileal cycling B cells had the highest prediction to cycling B cells in human ileum and germinal center dark zone (GC DZ) B cells in murine ileum, whereas activated B cells instead had highest the prediction to cells labeled as B cells in the human ileum and germinal center light zone (GC LZ) or resting B cells in murine ileum (Figs 3D, S20, and S21). A subset of cycling B cells had higher prediction scores to porcine peripheral T/ILC lineage lymphocytes and were more specifically predicted to be CD8αβ⁺ αβ T cells (Fig S19). Of all B/ASC types, porcine ileal activated B cells had the lowest average mapping scores to all reference datasets (range of means 0.530–0.692), suggesting lack of a similar cell population in porcine circulation and human or murine ileum (Figs 3C and S18).

## B lineage and cycling lymphocytes enriched in the ileum containing Peyer's patches

Because Peyer's patches are niches of GALT with specialized cellular functions different from those performed by cells in the lamina propria or epithelium, we assessed the impact of inclusion versus exclusion of Peyer's patches on cellular compositions recovered from the porcine ileum. As already shown in Figs 1A and S1B, ileal tissue was dissected into sections with Peyer's patches (PP), without Peyer's patches (non-PP), and a whole cross section of ileum (whole ileum) for cell isolation and scRNA-seq. At pseudo-bulk RNA-seq rather than scRNA-seq resolution, overall gene expression profiles of PP and the whole ileum were distinct from non-PP samples both before and after data quality control/filtering (Fig S22A). Analysis at single-cell resolution revealed similar results, whereby cell type proportions and overall cell numbers in whole ileum samples more closely resembled PP than non-PP samples (Figs 4A and S22B and C). At the cell lineage level, whole ileum and PP samples were composed primarily of B lineage lymphocytes (59.12% and 63.89%, respectively), followed by T/ILC lineage lymphocytes (38.13% and 33.17%, respectively; Fig 4B). In contrast, most cells from non-PP samples were T/ILC lineage lymphocytes (82.07%), and only 11.45% were B lineage lymphocytes (Fig 4B).

The presence and abundance of selected lymphocyte populations in different ileal regions were further validated ex vivo and in situ. Flow cytometry was used to assess B cell abundance via intracellular CD79α protein expression. Larger proportions of CD45⁺ leukocytes were CD79α⁺ in PP and whole ileum samples when compared with non-PP samples (Figs 4C and S23A). Immunohistochemistry (IHC) labeling revealed CD79α protein primarily in follicular areas of Peyer's patches but largely absent in lamina propria and epithelium (Fig 4D), indicating minimal CD79α detected in regions representative of non-PP samples. Because a dependable marker has not yet been established to identify ILCs in pigs, CD3ε protein staining was performed to label only T cells. By flow cytometry, higher percentages of CD3ε⁺ cells were detected within total CD45⁺ leukocyte populations of non-PP samples compared with PP or whole ileum samples (Figs 4E and S23A). By IHC, CD3ε protein staining was abundant in lamina propria, epithelium, and T

cell areas of Peyer's patches (Fig 4F), indicating CD3ε was present in regions representative of all ileal sections (PP, non-PP, and whole ileum). Collectively, ex vivo and in situ staining for CD79α and CD3ε protein supported scRNA-seq observations: B cells comprised a larger proportion of cells in PP and whole ileum samples, whereas T cells comprised a larger proportion of cells in non-PP samples. These results are informative in deciding sample preparation for inclusion of cells relevant to biological questions under investigation.

We further validated proportions of CD4 αβ, CD8 αβ, and γδ T cells in various regions of the ileum using flow cytometry and RNA in situ hybridization (ISH). T cells recovered via scRNA-seq were regrouped into CD4 αβ, CD8 αβ, and γδ T cells, and percentages of each subset within total T cells were calculated for each ileal scRNA-seq sample. Analysis revealed (1) increased proportions of CD4 αβ T cells in PP versus non-PP samples; (2) increased proportions of CD8 αβ T cells and γδ T cells in non-PP versus PP samples; and (3) intermediate proportions of all three T cell subsets in the whole ileum compared with PP and non-PP samples (Fig 4G). By flow cytometry, T cell proportions mirrored patterns obtained from scRNA-seq (Figs 4H and S23B). RNA ISH staining in regions of the ileum without Peyer's patches (Fig 4I, right) revealed *TRDC* (γδ T cells) and *CD8B* (CD8 αβ T cells) transcripts were primarily expressed within the epithelial layer, whereas *CD4* (CD4 αβ T cells) was expressed primarily within the lamina propria, supporting the conclusion that most γδ and CD8 αβ T cells were intraepithelial, and most CD4 αβ T cells resided in the lamina propria. Localization of *CD4*, *CD8B*, and *TRDC* did not change in epithelium and lamina propria adjacent to Peyer's patches. However, all three transcripts were also expressed by cells in the T cell zones of Peyer's patches, which were removed from non-PP samples (Fig 4I, left). Flow cytometry staining of epithelium-enriched, subepithelium-enriched, and merged cell fractions from PP and non-PP ileal samples was performed to validate ISH findings. The results revealed epithelium-enriched fractions from both PP and non-PP samples had higher percentages of γδ and CD8 αβ T cells and lower percentages of CD4 αβ T cells compared with subepithelium-enriched cell fractions (Fig S23C). In all, ex vivo and in situ staining to identify location of CD4 αβ, CD8 αβ, and γδ T cells mirrored results from scRNA-seq and provided further locational context of T cells in ileal epithelium, lamina propria, and Peyer's patches.

PP and non-PP samples were dissected by complete inclusion or exclusion of Peyer's patches, respectively, and could thus be directly compared to identify annotated cell types enriched in the presence versus absence of Peyer's patches. To identify cell type enrichment, cell neighborhoods (conglomerates of cells located near each other in the multidimensional space of the dataset) were identified from cells of PP and non-PP samples, and differential abundance analysis was performed on cell neighborhoods (Figs 4J and S24). Though T/ILC lineage lymphocytes comprised a greater proportion of total cells in non-PP than PP samples (Fig 4B), several T/ILC types were more abundant in PP samples, including cycling CD4 αβ T cells, cycling CD8 αβ T cells, and follicular CD4 αβ T cells, with at least 87.5% of cell neighborhoods significantly more abundant in PP samples for each cell type (Fig 4J). In contrast, cytotoxic and non-naive γδ T, CD8 αβ T, and group 1 ILCs, along with *SELL*ʰⁱ γδ T cells, had most of the cell neighborhoods (>50% for each cell type) significantly enriched in non-PP samples (Fig 4J).

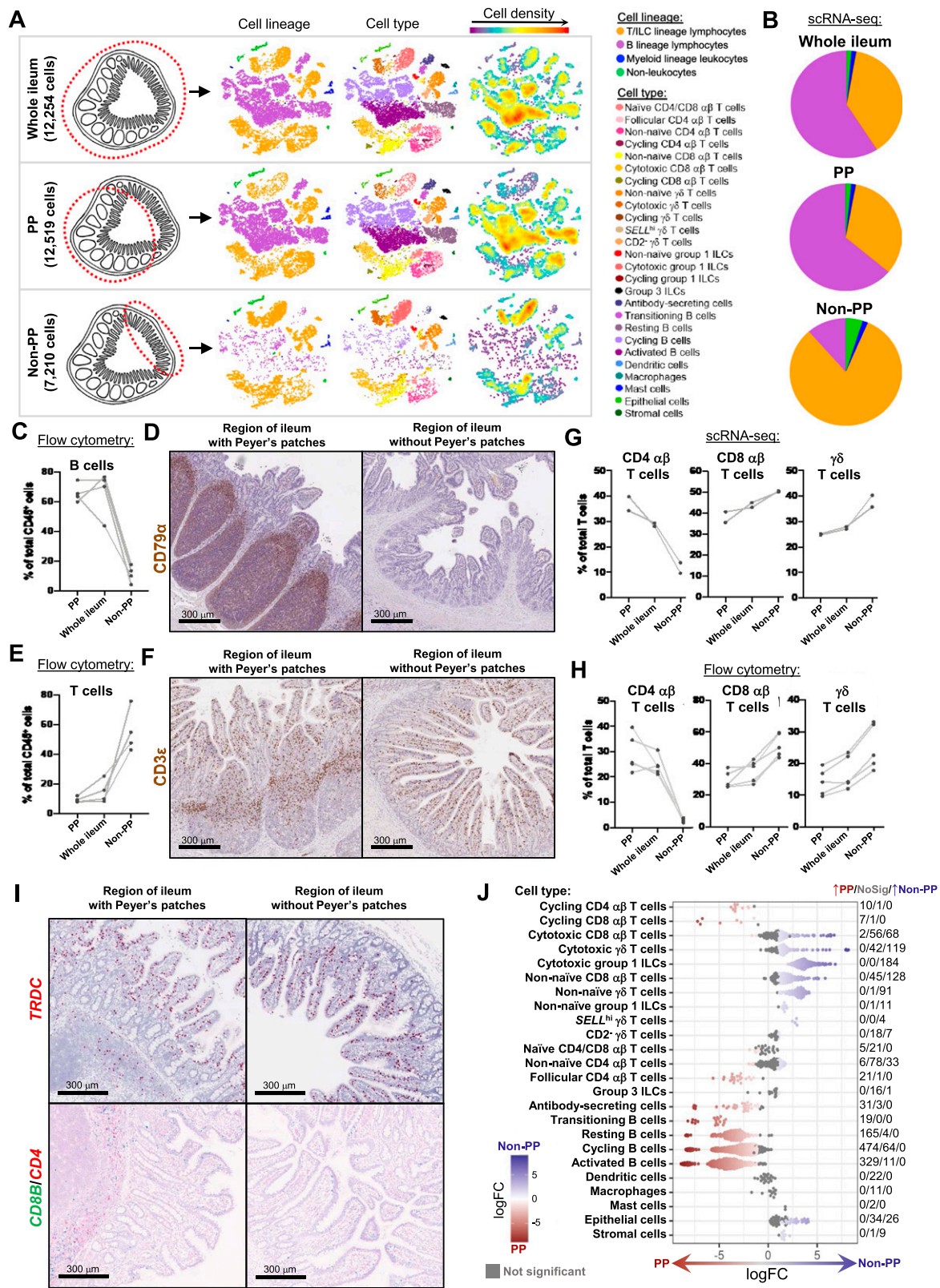

**Figure 4. Compositional differences in lymphocytes from the ileum with or without Peyer's patches.**
**(A)** Cell compositions of scRNA-seq data from the whole ileum (top), PP (middle), and non-PP (bottom) samples. Cells from each sample type (depicted on the far left) were combined from a total of two animals and overlaid onto t-SNE coordinates originally presented in Fig 1B–D. The total numbers of cells derived from the total of two animals for each sample type are listed on the far left. On the t-SNE plots, each point represents a single cell; the color of each point corresponds to cell lineage (left t-SNE),

Remaining T/ILC types had no or lower percentages of differentially abundant cell neighborhoods: 94.1% of group 3 ILC cell neighborhoods were not significantly differentially abundant; CD2⁻ γδ T cells had 28.0% of cell neighborhoods significantly increased in non-PP samples; naive CD4/CD8 αβ T cells had 19.2% of cell neighborhoods significantly increased in PP samples; and non-naive CD4 αβ T cells had 5.1% and 28.2% of cell neighborhoods significantly enriched in PP and non-PP samples, respectively (Fig 4J). Cycling γδ T and group 1 ILCs did not have any cell neighborhoods recovered for the analysis; however, no cycling group 1 ILCs were recovered from non-PP samples, and cycling γδ T cells derived from PP samples outnumbered those derived from non-PP samples by sixteen-to-three (Fig S22C). Similar to results evaluating compositions at the cell lineage level, all cell types from the B lymphocyte lineage (ASCs, transitioning B, resting B, cycling B, and activated B) had at least 88.1% of cell neighborhoods significantly more abundant in PP samples, indicating enrichment within Peyer's patches (Fig 4J). Collectively, differential abundance analysis indicated that B cells, ASCs, cycling T/ILCs, and follicular CD4 αβ T cells were more abundant in PP samples, likely because of association of these cell types with functions of ileal Peyer's patches and germinal center responses. Cytotoxic and non-naive subsets of γδ T, CD8 αβ T, and group 1 ILCs were more abundant in non-PP samples, indicating location and functions within the epithelium and/or lamina propria rather than in Peyer's patches of the ileum.

### Differential location of group 1 ILCs and group 3 ILCs in the porcine ileum indicated via ex vivo and in situ detection

To date, NK cells are the only porcine non–B/non–T lymphocyte subset identified and are typically identified as CD3ε⁻CD8α⁺ lymphocytes using ex vivo flow cytometry assessment (reviewed by Gerner et al [2009]). NK cell frequency in the porcine intestine has been assessed, but very few NK cells are detected in the porcine ileum (Sinkora et al, 2011; Annamalai et al, 2015, 2019; Potockova et al, 2015; Wasowicz et al, 2018). In contrast, we identified 2,594 cells (8.1% of total cells) as ILCs in porcine ileum via scRNA-seq, and few ILCs expressed CD8A (Figs 2B and S25). We therefore used scRNA-seq data to inform an approach to identify ILCs (not just CD3ε⁻CD8α⁺ NK cells) by flow cytometry.

Further query of ILC single-cell gene expression profiles revealed ILCs expressed CD2 but lacked gene expression of canonical T, B, and myeloid lineage leukocyte markers (CD3E, CD79A, and SIRPA* [encoding CD172α], respectively; Figs 2B and S25). Because porcine-specific immunoreagents are commercially available for CD2, CD3ε, CD79α, and CD172α, we identified cells corresponding to an ILC phenotype from the porcine ileum via flow cytometry as CD2⁺Lin⁻ cells (Lin⁻ = CD3ε⁻CD79α⁻CD172α⁻; Fig 5A). Nearly all CD2⁺Lin⁻ cells had forward- and side-scatter properties indicative of lymphocytes (Fig 5B). Most ILCs also expressed PTPRC (encoding CD45, considered to be a pan-leukocyte marker in pigs [reviewed by Piriou-Guzylack and Salmon (2008)]) in our scRNA-seq data. However, fewer group 3 ILCs expressed PTPRC (67.68%; Fig S25), and previous work in mice indicates CD45 expression may be lost by ILC3s, rendering it an inconsistent ILC marker (Xu et al, 2019). Congruently, the majority (≥83.1%) but not all CD2⁺Lin⁻ cells in the porcine ileum were CD45⁺ (Fig 5C).

To assess spatial context of ILCs in tissue and cautiously infer cell function based on location, we developed in situ methods for detection of porcine group 1 and group 3 ILCs in the ileum. From initial scRNA-seq analysis, activated, cytotoxic, and cycling group 1 ILCs all had high expression of ITGAE, encoding the

---

cell type (center t-SNE), or cell density (right t-SNE). **(B)** Pie charts showing proportions of cells from each annotated cell lineage within total cells derived from each sample type in (A). The color of a pie slice indicates cell lineage. The total area of each pie chart is not proportional to the total number of cells derived from each sample type. Proportions were calculated from total cells derived from two pigs for each sample type. **(C)** Plot of the percentage of B cells (CD79α⁺) within total leukocytes (CD45⁺; y-axis) from samples of the whole ileum, PP, and non-PP (x-axis), as assessed by flow cytometry gating shown in Fig S23A. Measurements from different sample types derived from the same animal are connected by a light gray line. **(D)** IHC staining for B cell CD79α protein (brown) in a region of the ileum with Peyer's patches (left) or without Peyer's patches (right). **(E)** Plot of the percentage of T cells (CD3ε⁺) within total leukocytes (CD45⁺; y-axis) from samples of the whole ileum, PP, and non-PP (x-axis), as assessed by flow cytometry gating shown in Fig S23A. Measurements from different sample types derived from the same animal are connected by a light gray line. **(F)** IHC staining for T cell CD3ε protein (brown) in a region of the ileum with Peyer's patches (left) or without Peyer's patches (right). **(G)** Plot of the percentage of CD4 αβ T cells (left), CD8 αβ T cells (center), or γδ T cells (right) within total T cells (y-axis) of the porcine-ileum scRNA-seq dataset. Percentages from samples of the whole ileum, PP, and non-PP are shown on the x-axis. CD4 αβ T cells included cells annotated as follicular CD4 αβ T cells, non-naive CD4 αβ T cells, or cycling CD4 αβ T cells and cells annotated as naive CD4/CD8 αβ T cells with prediction probability to porcine PBMC CD4⁺ αβ T cells > prediction probability to porcine PBMC CD8 αβ⁺ αβ T cells. CD8 αβ T cells included cells annotated as non-naive CD8 αβ T cells, cytotoxic CD8 αβ T cells, or cycling CD8 αβ T cells and cells annotated as naive CD4/CD8 αβ T cells with prediction probability to porcine PBMC CD8αβ⁺ αβ T cells > prediction probability to porcine PBMC CD4⁺ αβ T cells. γδ T cells included cells annotated as non-naive γδ T cells, cytotoxic γδ T cells, cycling γδ T cells, SELLʰⁱ γδ T cells, and CD2⁻ γδ T cells. Measurements from different sample types derived from the same animal are connected by a light gray line. **(H)** Plot of the percentage of CD4 αβ T cells (γδTCR⁻CD4⁺; left), CD8 αβ T cells (γδTCR⁻CD8β⁺; center), or γδ T cells (γδTCR⁺; right) within total T cells (CD3ε⁺; y-axis) from samples of the whole ileum, PP, and non-PP (x-axis), as assessed by flow cytometry gating shown in Fig S23B. Measurements from different sample types derived from the same animal are connected by a light gray line. **(I)** RNA ISH staining for TRDC (top, red), CD8B (bottom, green), or CD4 (bottom, red) transcripts in regions of the ileum with Peyer's patches (left) or regions of the ileum without Peyer's patches (right). **(J)** Differential abundance analysis of cell types from porcine-ileum scRNA-seq PP versus non-PP samples. Annotated cell types are listed on the y-axis. Each point represents an individual cell neighborhood, where a neighborhood was assigned as a specific cell type if >70% of cells within the neighborhood belonged to the specified cell type annotation. Cell neighborhoods with <70% of cells belonging to a single cell type are not shown. Gray points indicate cell neighborhoods that were not significantly more abundant in a specific sample type. Non-gray points indicate cell neighborhoods exhibiting differential abundance (P < 0.1), and red/blue fill of differentially abundant points corresponds to the magnitude and direction of logFC (also corresponding to values listed on the x-axis). Red indicates increased abundance in PP samples, whereas blue indicates increased abundance in non-PP samples. On the far right, counts of cell neighborhoods with increased abundance in PP samples/no differential abundance/increased abundance in non-PP samples are shown for each cell type. Cycling γδ T cells and cycling group 1 ILCs are not shown on the y-axis because of no cell neighborhoods being assigned to these cell types. scRNA-seq data shown in (A, B, G, J) were derived from the ileum of two 7-wk-old pigs. **(I)** Images shown in (I) were also taken from a 7-wk-old pig used for ileum scRNA-seq. Flow cytometry and IHC experiments were not performed on animals used for scRNA-seq. Flow cytometry experiments shown in (C, E) were conducted using four 6-wk-old pigs. Flow cytometry data shown in (H) was performed using five 9-wk-old pigs. IHC staining in (D) and (F) was completed on 6-wk-old pigs. Abbreviations: IHC, immunohistochemistry; ILC, innate lymphoid cell; ISH, in situ hybridization; logFC, log fold-change; NoSig, no significance; PBMC, peripheral blood mononuclear cell; PP, Peyer's patch; scRNA-seq, single-cell RNA sequencing; t-SNE, t-distributed stochastic neighbor embedding; TCR, T cell receptor.

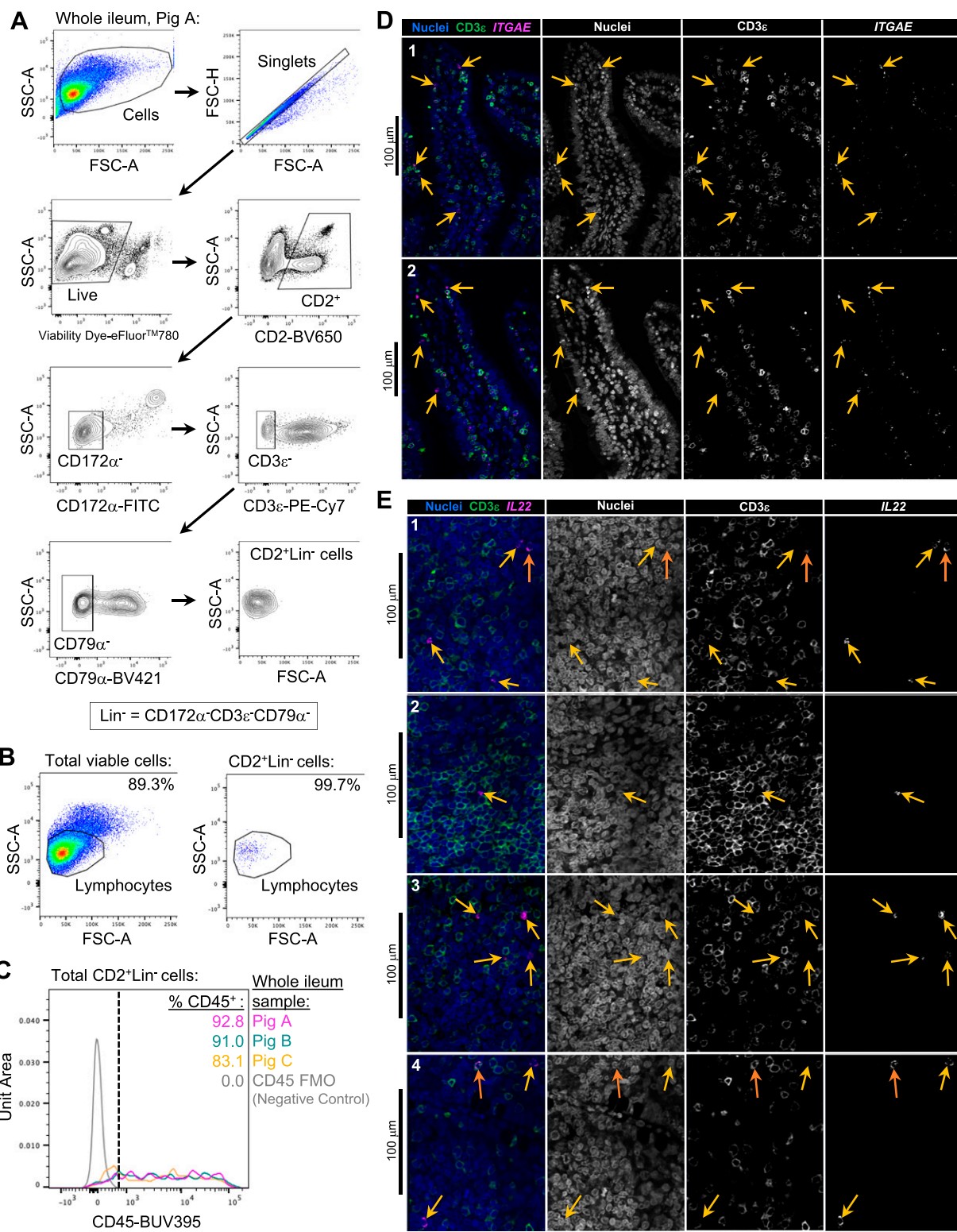

**Figure 5. Ex vivo and in situ identification of ILCs in the porcine ileum.**
**(A)** Flow cytometry gating strategy used to identify CD2⁺Lin⁻ (Lin⁻ = CD172α⁻CD3ε⁻CD79α⁻) within total viable cells of porcine ileal samples. **(C)** Gating is shown for a whole-ileum sample (containing both regions with and without Peyer's patches) for pig A (corresponding to pig IDs in (C)). **(B)** Flow cytometry forward- and side-scatter plots of total viable cells (left) and CD2⁺Lin⁻ cells (right) within a sample of the porcine whole ileum shown in (A). A gate identifying cells with scatter profiles consistent with lymphocytes is shown, with percentages of cells within the lymphocyte gate listed in the top right of each plot. **(C)** Histogram of the percentage of CD45⁺ cells within CD2⁺Lin⁻ cells identified from samples of the porcine ileum using the flow cytometry gating strategy shown in (A). A fluorescence-minus-one (FMO) sample lacking α-CD45

integrin *α* subunit of CD103 that is highly expressed by intraepithelial lymphocytes in the intestine of other species (Figs 2B and S25B and C; Kilshaw & Murant, 1990; Cepek et al, 1994; Mayassi & Jabri, 2018; Olivares-Villagómez & Van Kaer, 2018). Correspondingly, *ITGAE* was primarily detected in the epithelial layer of the porcine ileum by in situ analysis (Fig S26A). *ITGAE* was also strongly expressed by some T cells in our scRNA-seq dataset (Figs 2B and S25B and C); thus, we additionally assessed protein expression of CD3ε to decipher intraepithelial T cells (*ITGAE*⁺CD3ε⁺) from intraepithelial group 1 ILCs (*ITGAE*⁺CD3ε⁻). Dual *ITGAE*/CD3ε in situ labeling revealed most of the *ITGAE*⁺ intraepithelial cells were CD3ε⁺ T cells; however, *ITGAE*⁺CD3ε⁻ intraepithelial cells were also noted in the porcine ileum (Figs 5D and S26A). Though *ITGAE*⁺CD3ε⁻ intraepithelial cells could also be intraepithelial DCs, we found few DCs expressed *ITGAE* (Fig S25C), and *ITGAE*⁺CD3ε⁻ cells could also be found located in the apical epithelium that was in closest proximity to the lumen, whereas intraepithelial DCs reside only in the basement membrane (Farache et al, 2013). Thus, most of the *ITGAE*⁺CD3ε⁻ intraepithelial cells were assumed to be intraepithelial group 1 ILCs (Figs 5D and S26A).

Group 3 ILCs in the porcine ileum had high expression of *IL22* (Figs 2B and S25B and C), and *IL22* was detected in situ primarily in the lamina propria and T cell zones of Peyer's patches in porcine ileum (Fig S26B). Furthermore, most but not all *IL22*⁺ cells detected in situ lacked CD3ε expression, indicating that most of the *IL22*-expressing cells were not T cells (Figs 5E and S26B). Therefore, we identified group 3 ILCs as *IL22*⁺CD3ε⁻ cells, though the in situ staining combination also identified a minority of *IL22*⁺CD3ε⁺ (presumably T$_{H22}$) cells. Collectively, query of gene expression for ILCs at single-cell resolution allowed us to develop new reagent panels for ex vivo and in situ analysis of ILCs in the porcine ileum that expanded on previous panels catering to only NK cell identification.

### Transcriptional distinctions between porcine ileal and circulating ILCs

Porcine NK cells were not representative of porcine ileal ILCs; therefore, ileal ILCs required different markers for ex vivo and in situ detection (Figs 5, S25, and S26). As porcine peripheral NK cells are readily characterized, we wanted to further explore transcriptional distinctions between porcine ileal ILCs and porcine peripheral ILCs (i.e., NK cells) to infer potential cell functions and identify additional targets for ex vivo and in situ detection. PBMCs from the same two pigs were processed for scRNA-seq in parallel to

ileal samples, and peripheral ILCs were identified by the same criteria used to identify ILCs in the ileum (refer to the Materials and Methods section for full details). Porcine peripheral ILC gene expression profiles were largely concordant with NK cells described in our previous work (Fig S27A–E and Table S10; Herrera-Uribe et al, 2021). A new merged ILC scRNA-seq dataset containing only single cells annotated as ILCs from both PBMCs and ileal samples was generated (Fig 6A). Within the merged ILC dataset, 41.3% of detected cell neighborhoods were differentially abundant based on ileal versus peripheral cell origin (Fig 6B and C), suggesting differences between at least some ileal and peripheral ILCs may exist and meriting further exploration.

Multidimensional differential gene expression (DGE) was performed to identify differentially expressed genes (DEGs) independent of cell type annotations and sample origin of cells (Table S10). Hierarchical clustering of DEGs with the lowest *P*-values defined nine gene modules with varying patterns of detection (Figs 6D and E and S28A–C). Gene module 4 had highest detection in peripheral ILCs, whereas module 5 had higher detection across all four types of ileal ILCs (Fig 6D and E), allowing us to easily segregate ILCs by peripheral or ileal origin based on gene module 4 versus 5 detection scores (Fig 6F). Gene module 7 had highest detection in ileal group 1 ILCs (activated, cytotoxic, and cycling), whereas gene modules 3 and 8 detection scores correlated positively with one another and were higher in ileal group 3 ILCs (Figs 6D and E and S28D). Collectively, the differences noted allowed for segregation between ileal group 1 and group 3 ILCs on the basis of detection for gene module 7 and 3/8, respectively (Fig 6G and H). Therefore, gene module 4 was considered as highly specific to peripheral ILCs, gene module 5 to all ileal ILCs, gene module 7 to ileal group 1 ILCs, and gene modules 3 and 8 to ileal group 3 ILCs. Further refinement focused on genes with the highest specificity for respectively assigned ILC subsets from gene modules in Fig 6D and E, yielding core gene signatures for peripheral ILCs, all ileal ILCs, ileal group 1 ILCs, and ileal group 3 ILCs (Fig 7A–D).

An 86-gene signature derived from gene module 4 was identified for peripheral ILCs (Fig 7A) and included several genes recognized as canonical markers for porcine NK cells (*CD8A*, *FCGR3A\**, *ITGAM*, *NCR3\**, and *HCST*; Denyer et al, 2006; Piriou-Guzylack & Salmon, 2008; Gerner et al, 2009; Toka et al, 2009; Mair et al, 2013) or recognized as canonical NK cell markers across other species (*KLRG1\**, *CD99*, *TBX21*, and *GZMM*; Townsend et al, 2004; Huntington et al, 2007; Knox et al, 2014; Robinette et al, 2015; Crinier et al, 2018). *FCGR3A\** and *CD8A* encode for CD16 and CD8α, respectively, which are the two primary markers used to identify porcine NK cells at the protein level (reviewed by Piriou-

antibody staining was used as a negative control. **(D)** Dual fluorescent staining of CD3ε protein and *ITGAE* RNA in villi (epithelium + lamina propria) of the porcine ileum. Left column: overlay of all stains, including CD3ε protein (green), *ITGAE* RNA (magenta), and nuclei (DAPI staining; blue). Additional columns show individual stain overlays in white, including (from left to right) nuclei, CD3ε protein, and *ITGAE* RNA. Panels of two separate villi are shown in each row. Panels were selected from larger stitched images as shown in Fig S26A. Yellow arrows indicate location of *ITGAE*⁺CD3ε⁻ cells. **(E)** Dual fluorescent staining of CD3ε protein and *IL22* RNA in lamina propria/GALT of the porcine ileum. Left column: overlay of all stains, including CD3ε protein (green), *IL22* RNA (magenta), and nuclei (DAPI staining; blue). Additional columns show individual stain overlays in white, including (from left to right) nuclei, CD3ε protein, and *IL22* RNA. Panels of four separate tissue locations are shown in each row. Panels were selected from larger stitched images as shown in Fig S26B. Yellow arrows indicate location of *IL22*⁺CD3ε⁻ cells. Orange arrows indicate location of *IL22*⁺CD3ε⁺ cells. Flow cytometry experiments shown in (A, B, C) were conducted using three 6-wk-old pigs. Dual IF/ISH experiments shown in (D, E) were conducted using a 7-wk-old pig used for ileum scRNA-seq. Abbreviations: FMO, fluorescence-minus-one; FSC-A, forward scatter area; FSC-H, forward scatter height; GALT, gut-associated lymphoid tissue; IF, immunofluorescence; ISH, in situ hybridization; ILC, innate lymphoid cell; scRNA-seq, single-cell RNA sequencing; SSC-A, side scatter area.

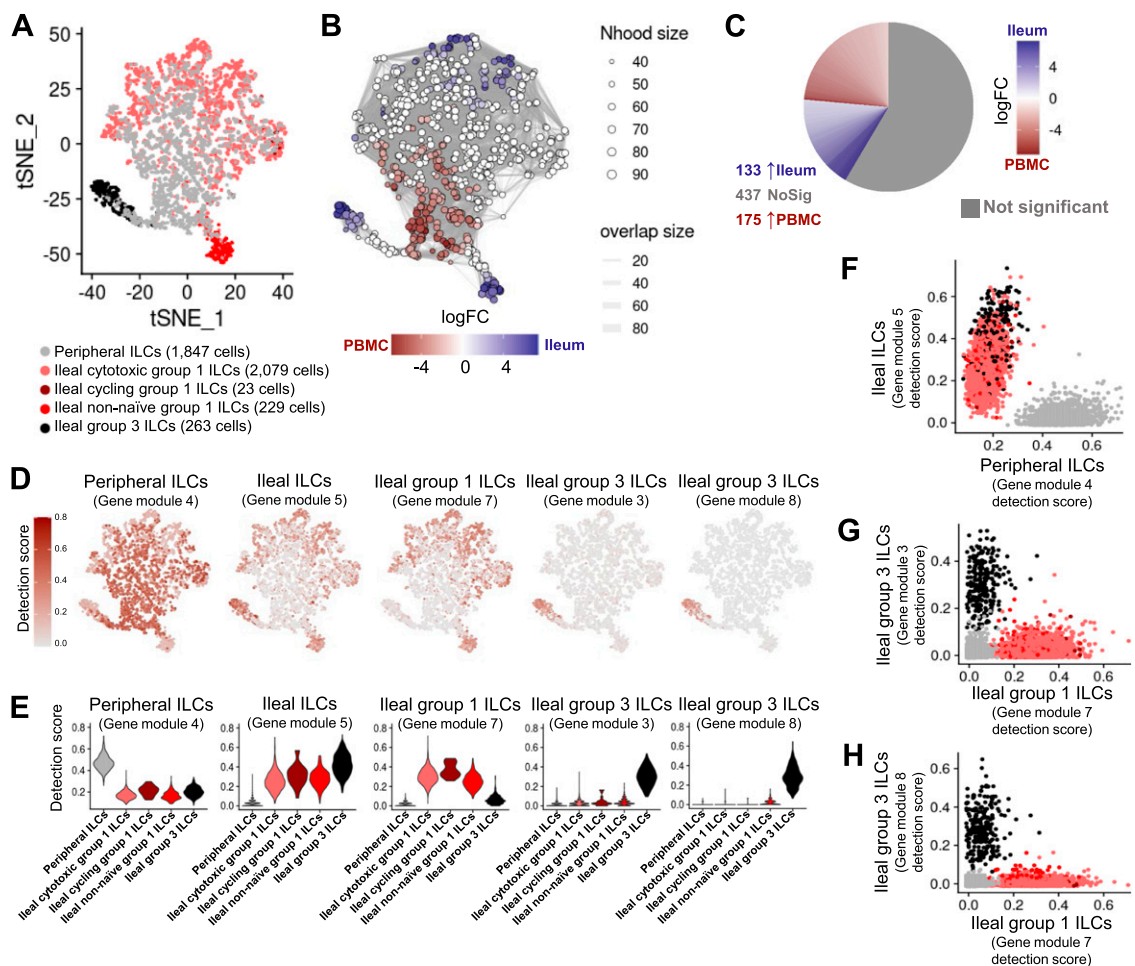

**Figure 6. Peripheral ILCs are transcriptionally distinct from ileal ILCs.**
**(A)** Two-dimensional t-SNE visualization of 4,441 cells recovered from the porcine ileum (2,594 cells) and porcine PBMCs (1,847 cells) via scRNA-seq and classified as ILCs in Figs S5C–F and S27D and E. Each point represents a single cell; color of a point corresponds to one of five ILC annotations. The number of cells belonging to each cell type is listed in the color key below. Derivation from the ileum or PBMC scRNA-seq samples for annotated ILC types is indicated in the annotation name. **(B)** Cell neighborhoods identified by differential abundance analysis between cells derived from the ileum and PBMCs as shown in (A), overlaid onto t-SNE coordinates also shown in (A). Size of a circle indicates the number of cells in a neighborhood (Nhood size); color of a circle indicates magnitude of logFC in abundance in the ileum (blue) versus PBMCs (red); width of lines between cell neighborhoods indicates the number of overlapping cells found in each of two neighborhoods (overlap size). **(C)** Pie chart of differential abundance results for cell neighborhoods shown in (B). Gray indicates the proportion of cell neighborhoods that were not differentially abundant, whereas cell neighborhoods with significantly increased abundance (P < 0.01) in the ileum or PBMC samples are shown in blue and red, respectively. The logFC magnitude of differential abundance is also shown by red or blue shading. **(D)** Selected gene module detection scores from multidimensional differential gene expression analysis of cells shown in (A) overlaid onto two-dimensional t-SNE visualization coordinates. Color of a point corresponds to detection score for a gene module within a cell. **(E)** Violin plots summarizing gene module detections scores shown in (D) (y-axis) across annotated ILC types shown in (A) (x-axis). **(F)** Scatter plot of gene module 5 detection scores (y-axis) versus gene module 4 detection scores (x-axis) for all cells shown in (A). Each point represents a single cell; color of a point corresponds to cell type annotations shown in (A). **(G)** Scatter plot of gene module 3 detection scores (y-axis) versus gene module 7 detection scores (x-axis) for all cells shown in (A). Each point represents a single cell; color of a point corresponds to cell type annotations shown in (A). **(H)** Scatter plot of gene module 8 detection scores (y-axis) versus gene module 7 detection scores (x-axis) for all cells shown in (A). Each point represents a single cell; color of a point corresponds to cell type annotations shown in (A). scRNA-seq data shown in (A, B, C, D, E, F, G, H) were derived from the ileum and PBMCs of two 7-wk-old pigs. Ileum and PBMC samples for scRNA-seq were collected from the same two pigs and processed in parallel. Abbreviations: ILC, innate lymphoid cell; logFC, log fold-change; Nhood, neighborhood; No Sig, no significance; PBMC, peripheral blood mononuclear cell; scRNA-seq, single-cell RNA sequencing; t-SNE, t-distributed stochastic neighbor embedding.

Guzylack and Salmon [2008] and Gerner et al [2009]). Thus, inclusion of *FCGR3A*\* and *CD8A* in a gene signature for peripheral ILCs concordant with NK cell descriptions suggests current porcine NK cell identifiers are insufficient for pan-ILC identification in pigs. Additional peripheral ILC signature genes encoded for integrins (*ITGB2, ITGA4, ITGAM,* and *ITGAL*) and other molecules associated with cell receptors/signaling (*FGR, PTPRC, CX3CR1,* and *S1PR5*), consistent with enrichment of biological processes including

integrin-mediated signaling pathway (*GO:0007229*), leukocyte cell–cell adhesion (*GO:0007159*), and receptor clustering (*GO:0043113*) in gene module 4.

Thirty genes derived from gene module 5 comprised an ileal ILC gene signature identifying both group 1 and group 3 ILCs in the porcine ileum (Fig 7B). The gene signature for ileal ILCs included *CD69*, encoding for the cell surface receptor CD69 used to identify tissue resident and recently activated cells (Shiow et al, 2006;

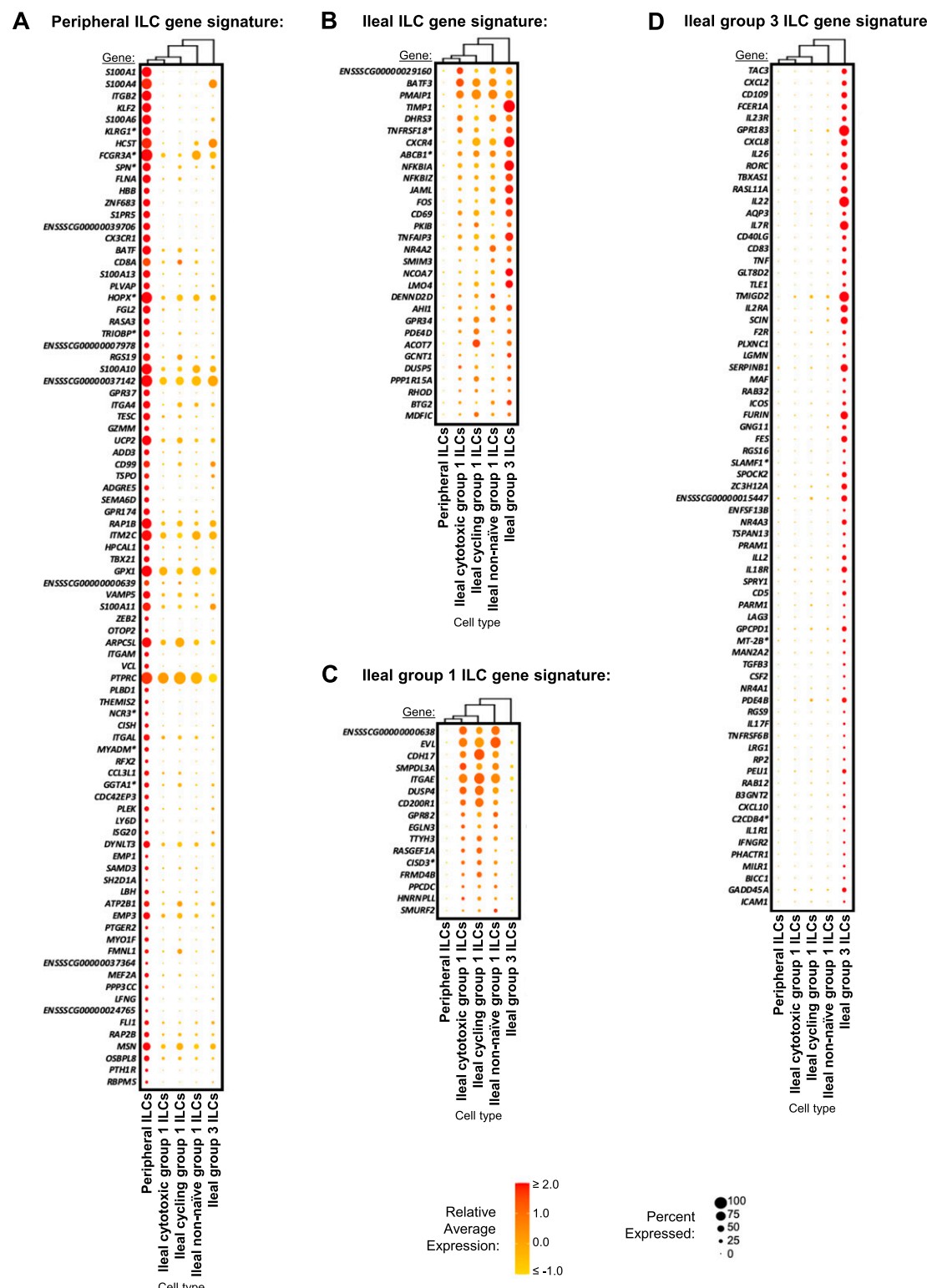

**Figure 7. Core gene signatures of ileal and peripheral ILCs.**
**(A, B, C, D)** Dot plots showing expression patterns within each annotated ILC type for selected genes from gene module 4 used to create a peripheral ILC gene signature (A), gene module 5 used to create an ileal ILC gene signature (B), gene module 7 used to create an ileal group 1 ILC gene signature (C), and gene modules 3 and 8 used to create an ileal group 3 ILC gene signature (D). Genes are shown on the y-axis, and annotated ILC types from the porcine ileum and PBMCs are shown on the x-axis. Within the dot plot, size of a dot corresponds to the percentage of cells expressing a gene within an annotated cell type; color of a dot corresponds to average expression level of a gene for those cells expressing it within a cell type relative to all other cells in the dataset shown in Fig 6A. Hierarchical relationships of annotated ILC types are shown

Ashouri & Weiss, 2017; Szabo et al, 2019b). Enriched biological pathways for module 5 included those associated with cell metabolism, including positive regulation of macromolecule metabolic process (*GO:0010604*), positive regulation of cellular protein catabolic process (*GO:1903364*), and long-chain fatty-acid metabolic process (*GO:0001676*), suggesting that ileal ILCs were in a highly active metabolic state.

A gene signature for ileal group 1 ILCs included 16 genes derived from gene module 7, as shown in Fig 7C. Similar to gene module 5, gene module 7 was enriched for metabolic processes, including catabolic processes such as nucleoside phosphate catabolic process (*GO:1901292*) and biosynthetic processes such as purine-containing compound biosynthetic process (*GO:0072522*), suggesting an active metabolic state defining group 1 ILCs in the ileum. Notably, *ITGAE*, which we used to identify intraepithelial group 1 ILCs in Fig 5D, was found in the ileal group 1 ILC gene signature, further supporting its use as an effective group 1 ILC marker in conjunction with other key markers in the porcine ileum.

Group 3 ILCs had a core gene signature of 71 genes derived from gene modules 3 and 8 (Fig 7D). Many of the signature genes were associated with a type 3 immune response and largely overlapped with DEGs identified for ileal group 3 ILCs relative to all other ileal T/ILCs and relative to all other ileal ILCs (Tables S4 and S8). Similarly, enriched biological processes for gene modules 3/8 also had high overlap with enriched biological processes found from DEGs in Tables S4 and S8. Notably, *IL22*, used as an in situ marker for group 3 ILCs in Fig 5E, was detected in the ileal group 3 ILC gene signature, supporting its use as an effective group 3 ILC marker in conjunction with other key markers, not only relative to other cells in ileum but also relative to peripheral ILCs.

## Discussion

Understanding intestinal lymphocyte identity and function to cumulatively promote intestinal health outcomes in pigs has implications for biomedical research, animal health, and global food security. Phenotyping of porcine lymphocytes based on protein expression of a handful of cell surface markers has provided limited information about the biological dynamics of intestinal lymphocytes. Our work provides greater biological insight through further characterization of ileal lymphocytes into inferred functional subsets at transcriptional resolution not previously achieved in pigs, and we also include cross-species/cross-tissue comparisons and spatial context of specific lymphocyte types in different regions of the ileum. In addition to discoveries described herein, the data serves as a resource to be further explored and dissected for information about the heterogeneous landscape of lymphocytes in the porcine ileum.

Pigs readily serve as a biomedical model, with several innovations already applied to study intestinal lymphocytes. For example,

germ-free pigs are used to study the dependence of intestinal lymphocyte development on microbial colonization (reviewed by Butler et al [2009], Sinkora and Butler [2009, 2016], and Pabst [2020]). Germ-free pigs can be colonized with human microbiomes to generate microbially humanized pigs for understanding the impact of the microbiota on local cellular responses associated with mucosal vaccination strategies and subsequent protection (Kumar et al, 2018; Miyazaki et al, 2018). Moreover, pigs serve as models for studying the impact of stress, nutrition, and infectious disease on intestinal physiology and immune status (reviewed by Gonzalez et al [2015], Roura et al [2016], Ziegler et al [2016], Moeser et al [2017], Hryhorowicz et al [2020], and Käser [2021]). Bolstering lymphocyte-mediated intestinal immunity is important for both humans and pigs, but mechanistic approaches rely heavily on first understanding key lymphocytes in the porcine intestine and recognizing shared annotation with humans. To help overcome these obstacles, we identified and described biological implications of lymphocytes in the porcine ileum in comparison to both human and murine ileum. Collectively, our data indicate a general consensus of annotated lymphocyte types across species; however, a few notable differences were described. Therefore, our cross-species comparisons serve as a foundation for deeper resolved exploration of pigs as models for human medicine. We further explore parallels between pigs and humans as it relates to intestinal lymphocytes throughout the rest of the discussion. However, we also note that some interspecies differences might be attributed to technical variables, such as differences in tissue collection, cell isolation, sequencing, or data interpretation.

A caveat of scRNA-seq is loss of cellular spatial context when tissues are dissociated, which we partially overcame by differentially dissecting the ileum into sections with or without Peyer's patches before cell isolation. Cycling T/ILCs, follicular CD4 $\alpha\beta$ T cells, B cells, and ASCs were enriched in Peyer's patches, suggesting induction of germinal center antibody responses characteristic of GALT. Data support findings from human research, showing an enrichment of cycling cells, CD4 $\alpha\beta$ T cells, and B cells in Peyer's patches (Fujihashi et al, 2008; Junker et al, 2009) and thus strengthening the rationale for applicability of pigs to study Peyer's patch–associated immune induction in biomedical research. Non-cycling $\gamma\delta$ T, CD8 $\alpha\beta$ T, and group 1 ILCs were enriched in the absence of Peyer's patches and detected in situ primarily within the ileal epithelium, suggesting these three cell types mostly comprise the intraepithelial lymphocyte (IEL) community in the porcine ileum, similar to humans (reviewed by Lutter et al [2018], Mayassi and Jabri [2018], and Olivares-Villagómez and Van Kaer [2018]). Though porcine intraepithelial group 1 ILCs have not been studied in detail, we have conducted research investigating compositional changes of intraepithelial T lymphocytes (T-IELs) in the porcine intestine after weaning (Wiarda et al, 2020). Cross-species parallels were noted in our previous work, including a predominance of CD8$^+$ $\alpha\beta$ over $\gamma\delta$ T-IELs in the small intestine of pigs and humans (Jarry et al,

---

with a dendrogram on the top of each dot plot. scRNA-seq data shown in (A, B, C, D) were derived from the ileum and PBMCs of two 7-wk-old pigs. Ileum and PBMC samples for scRNA-seq were collected from the same two pigs and processed in parallel. *Ensembl identifiers found in gene annotation were converted to gene symbols; refer to methods section "*Gene name modifications*" for more details. Abbreviations: ILC, innate lymphoid cell; PBMC, peripheral blood mononuclear cell; scRNA-seq, single-cell RNA sequencing.

1990; Lundqvist et al, 1995; Olivares-Villagómez & Van Kaer, 2018; Wiarda et al, 2020). In contrast, γδ T-IELs are more frequent than CD8⁺ αβ T-IELs in the murine small intestine (Guy-Grand et al, 1991; Hoytema van Konijnenburg et al, 2017). Thus, our previous work gives initial support of pigs for IEL-centric biomedical research, and scRNA-seq of γδ T, CD8 αβ T, and group 1 ILCs presented herein may help to build on this application. We caution that lamina propria versus epithelial location of single cells cannot be definitively concluded solely from our scRNA-seq data. However, T cells studied via scRNA-seq from the lamina propria and epithelium of human ileum had highly overlapping transcriptional profiles (Jaeger et al, 2021), and the same may be true for pigs.

Non-naive T and B cells in the porcine ileum were transcriptionally different from comparable cells in blood, as indicated by lower average mapping scores to porcine PBMCs. Transcriptional differences between ileal- and blood-derived non-naive T/B cells likely occurred because of inherently activated and/or differentiating states of ileal lymphocytes because of exposure to luminal contents, including the microbiota and microbial-derived metabolites. Our supposition is reinforced by work in germ-free pigs, where small-intestinal non-naive T and B cell abundances are reduced in the absence of microbial exposure (Rothkötter & Pabst, 1989; Rothkötter et al, 1994; Barman et al, 1996; Haverson et al, 2007; Sinkora et al, 2011; Potockova et al, 2015). Differences in T/B cells from mucosal tissues versus blood are also observed in humans, where non-naive lymphocytes increase in abundance, and tissue-specific gene signatures for lymphocyte activation, tissue residency, and/or effector functions are observed for tonsillar B cells (Glass et al, 2020; King et al, 2021), lung-derived T cells (Szabo et al, 2019a), and intestinal T cells (Venema et al, 2019) compared with similar lymphocyte subsets in the periphery. Collectively, the aforementioned studies and our own work suggest mucosal sites (including the ileum) are locations for congregation of non-naive T/B cells transcriptionally distinct from T/B cells found in the periphery, likely due in large part to ongoing immune stimulation via microbial exposure and other context-dependent signals at mucosal surfaces. Though transcriptional reprogramming of T cells and ILCs upon recruitment into the intestinal tract likely contributes to transcriptional distinctions between some T cells and ILCs found in blood and intestine, other intestinal T cells and ILCs are likely unique to the intestine. For example, thymic precursors of some intestinal resident T cells are recruited directly to the intestine where they further mature as tissue resident cells (Poussier & Julium, 1994; Poussier et al, 1992; McDonald et al, 2014). Therefore, findings ultimately support two potential scenarios for transcriptional distinctions between non-naive ileal and peripheral T cells and ILCs: (1) the existence of distinct cell populations exclusive to either circulation or the intestine and (2) transcriptional reprogramming of T cells and ILCs upon recruitment into the intestinal tract.

Similar to intestinal T and B cells, ILCs in the porcine intestine are likely heavily influenced by the microbiota; however, little is known about the influence of microbial colonization on porcine intestinal ILCs because of lack of methods to clearly label and identify intestinal ILCs in pigs. We established gene expression profiles and locational context of group 1 and group 3 ILCs in the porcine ileum to cautiously make functional inferences about ILCs in the porcine

intestine. Our findings support comparable roles of porcine ileal ILC functions to intestinal ILCs in humans and mice, such as epithelial patrolling behaviors of intraepithelial group 1 ILCs (Fuchs et al, 2013; Talayero et al, 2016; Van Acker et al, 2017) and contribution of lamina propria and GALT-associated group 3 ILCs in immune defense, regulation of the commensal microbiota, GALT development, tissue homeostasis, and antibody production (Satoh-Takayama et al, 2008; Tsuji et al, 2008; Sonnenberg et al, 2012; Kruglov et al, 2013; Guo et al, 2014; Mortha et al, 2014; von Burg et al, 2014; Aparicio-Domingo et al, 2015). In mice, intestinal ILC1s proportionally contract and convert to a less polarized ILC3-like regulatory profile in germ-free compared with specific pathogen-free animals, suggesting ILC1 recruitment, differentiation, and/or function may be dampened without microbial stimulation (Gury-BenAri et al, 2016), and it remains plausible that similar phenomena occur in pigs. In support, porcine ileal ILCs gene signatures appeared to be heavily influenced by tissue-specific cell activation when compared with peripheral ILCs, and microbiota-derived signals likely supply tissue-specific transcriptional imprinting. In humans, ILCs from the tonsil, lung, and colon have tissue-specific gene expression indicative of tissue residency, cell activation, and modified metabolism compared with peripheral ILCs (Mazzurana et al, 2021), again supporting the contribution of external stimuli in driving activation state. Therefore, our data and newly developed methods for ILC detection should prove useful for further investigation, including defining the impact of the microbiota or intestinal infection on intestinal ILC function in pigs. We could not identify ileal group 2 ILCs in the porcine ileum; however, it is possible that group 2 ILCs were not found because of low abundances occurring under steady state conditions. In humans, intestinal group 2 ILCs are similarly rare under steady state conditions but may increase in abundance under conditions that facilitate their recruitment and/or expansion in the intestine, such as with parasitic infection of cancer (Meininger et al, 2020; Qi et al, 2021), and it remains possible that similar phenomena may occur in pigs. In contrast, group 2 ILCs comprise approximately one-quarter of ILCs in the lamina propria of the murine small intestine (Dutton et al, 2017). Thus, predominance of group 1 and group 3 ILCs in the pig ileum provides initial indication that pigs may have intestinal ILC populations more similar in composition to humans than that of mice under steady state.

Direct comparison of porcine ileal ILCs and peripheral NK cells brings into question the current methods for identification of non-T/non-B lymphocytes in pigs, as key markers used to identify NK cells (the only non–T/non–B lymphocyte subset currently identified in pigs) were found in gene signatures for peripheral but not ileal porcine ILCs. Ileal group 3 ILCs were easily distinguishable from peripheral NK cells, whereas ileal group 1 ILCs shared more similarities to peripheral NK cells. It remains undetermined whether ileal group 1 ILCs are ILC1s or NK cells, as detected differences between ileal group 1 ILCs and circulating NK cells in our data could indicate (1) identification of ILC1s present in the ileum but largely absent from periphery, (2) tissue-specific differences in ileal-derived NK cells that do not fit current phenotypic descriptions used to identify NK cells in pigs, or (3) a combination of both. Unfortunately, differentiation between ILC1s and NK cells remains complicated even in better characterized humans and mice

because of species-to-species and tissue-to-tissue peculiarities within what can still be considered relatively recently discovered immune cell subsets (Robinette et al, 2015; Simoni et al, 2017; Simoni & Newell, 2017; Van Acker et al, 2017; Meininger et al, 2020). Regardless, we stress the importance of focusing on cell function over technical phenotype classification, as the functional role of cells in the context of immune protection and immunopathology is ultimately what contributes to biological outcome. To this point, gene signatures obtained for ileal and peripheral ILCs may further be used for identifying biomarkers for detection of specific ILC subsets and understanding their biological functions.

To our knowledge, our findings encompass the first single-cell descriptions of the immune cell transcriptomic landscape in the porcine intestine, with primary focus on the study of lymphocytes in the ileum, including T cells, ILCs, B cells, and ASCs. In addition, non-lymphocyte cell types in the intestine play important functional roles in intestinal health but fell outside the scope of the current study. The diverse spectrum of biological states for cells captured via scRNA-seq is difficult to holistically describe, and we only scratch the surface of the biological information and complexities contained within our scRNA-seq data. Although our data may serve as a starting point for understanding the roles of specific immune cells in a specific biological scenario, such understandings are best fine-tuned using the most appropriate species, tissue, and biological scenario of interest. Consequently, our data serve as an exciting starting point for query to seed future research questions.

# Materials and Methods

## Animals and sample collection

All animal experiments were performed according to procedures approved by the Institutional Animal Care and Use Committee (IACUC) at Iowa State University or the National Animal Disease Center, Agricultural Research Service (ARS), United States Department of Agriculture (USDA). Mixed-breed and mixed-gender pigs were obtained from commercial nursery settings for all experiments and assays. All pigs were weaned at ~3 wk of age. For scRNA-seq, samples were collected from one female (pig 1) and one male (pig 2) at ~7 wk of age. Postmortem intestinal tissues collected to complete ex vivo and in situ assays were from control animals in unrelated studies to reduce the overall number of animals used. The ages of animals are listed within respective figure captions and ranged from ~5–9 wk of age. Humane euthanasia was carried out immediately preceding tissue collections.

## Cell isolations

### Ileum for scRNA-seq

For ileal cell isolations, reagents were equilibrated to RT, and samples were stored at RT between steps. Immediately after humane euthanasia, ileal tissue was collected and stored in stabilization buffer (2 mM EDTA [AM9260G; Invitrogen], 2 mM L-glutamine [25-030; Gibco], and 0.5% BSA [A9418; Sigma-Aldrich] in HBSS [14175; Gibco]) for transport back to the lab.

In the lab, the exterior muscularis layer was peeled off, and tissues were cut open longitudinally to expose the lumen. Tissues were gently rinsed with PBS, pH 7.2 (made in-house), to remove intestinal contents and carefully dissected into regions of interest (PP, non-PP, and whole ileum), as shown in Fig S1B. Dissected tissues were weighed out to between 1 and 1.5 g per sample for further use.

Tissues were sequentially transferred to and incubated in the following solutions in a shaking incubator (MaxQ 4000; Thermo Fisher Scientific) at 37°C, 200 rpm: 20 min in 30 ml mucus dissociation solution (5 mM dithiothretol [DTT; 15508; Invitrogen] and 2% heat-inactivated FCS [A38401; Gibco] in HBSS); 25 min in 30 ml epithelial removal solution (5 mM EDTA and 2% FCS in HBSS), repeated with fresh solution for a total of three times; and 10 min in 20 ml wash solution (10 mM Hepes [BP299; Fisher BioReagents] in HBSS). Epithelial removal and wash solutions were retained for processing of epithelial cells, whereas mucus dissociation solution was discarded.

After incubation in wash solution, tissues were minced and placed into gentleMACS C tubes (130-093-237; Miltenyi) containing 15 ml enzyme digestion solution (10 mM Hepes, 0.2 U/ml Liberase TM [5401127001; Roche], and 30 μg/ml DNase I [D5025; Sigma-Aldrich] in HBSS). Tissue dissociation was carried out using the intestine C-tube protocol on a gentleMACS Octo Dissociator (130-095-937; Miltenyi) followed by incubation at 37°C, 200 rpm for 45 min and another round of mechanical dissociation using the gentle-MACs intestine C-tube protocol. A total of 10 ml stabilization buffer was added to each C-tube, and contents were strained through sterile nonwoven surgical gauze sponges (GZNW44; Starryshine), followed by a 100-μm nylon mesh cell strainer (352360; BD Falcon).

Although tissues were incubating in enzyme digestion solution, epithelial cells were collected by passing epithelial removal and wash solutions through a 100-μm nylon mesh cell strainer, centrifuging at 450$g$ for 8 min RT, and resuspending in supplemented HBSS (2 mM L-glutamine and 2% FCS in HBSS).

Cell fractions from epithelial isolation and tissue digestion for each sample were combined, centrifuged at 450$g$ for 8 min RT, and resuspended in 24 ml of RT 70% Percoll (1.088 g/ml at 22°C; 17-0891-01; GE Healthcare Life Sciences). Aliquots of 8 ml cell/Percoll suspension were overlayed with 4 ml HBSS and centrifuged at 400$g$ for 30 min RT, with slow acceleration and centrifuge break turned off. The density interphase layer of cells was collected, washed with supplemented HBSS, centrifuged at 450$g$ for 8 min RT, and resuspended again in supplemented HBSS. Quantity and viability of cells was assessed by the Muse Count & Viability Assay Kit (MCH100102; Luminex) using a Muse Cell Analyzer (0500-3115; Luminex).

To further enrich for live cells, cell suspensions were passed through another 100-μm nylon filter, centrifuged at 300$g$ for 10 min RT, and processed using a Dead Cell Removal Kit (130-090-101; Miltenyi). Cells were resuspended in 100 μl kit microbeads per $10^7$ total cells, incubated for 15 min, and divided into four equal parts per sample that were each rinsed with 1 ml 1× Binding Buffer. A total of four separate LS columns (130-042-401; Miltenyi) were used for the divided samples to facilitate magnetic sorting using a Multi-MACS Cell24 Separator Plus (130-098-637; Miltenyi). LS columns were prerinsed with 1× Binding Buffer before applying cells.

Columns were rinsed with an additional 3 ml of 1× Binding Buffer twice to facilitate cell pass through. The negative cell pass through was collected, centrifuged 300*g* for 10 min RT, resuspended in supplemented HBSS, centrifuged again, and resuspended in a final volume of supplemented HBSS. Muse count and viability was again assessed and deemed adequate for scRNA-seq (>84% live cells per sample). Samples were transported to the sequencing facility (~15 min transport), and cell viability was reassessed using a Countess II Automated Cell Counter (Thermo Fisher Scientific). Viabilities from the Countess readings were deemed adequate to proceed with partitioning (>76% live cells per sample) for scRNA-seq.

### PBMCs for scRNA-seq
PBMCs were isolated using Cell Preparation Tubes (CPT; 362782; BD Biosciences) according to manufacturer's recommendations.

### Ileum for flow cytometry
Ileal cells collected for flow cytometry experiments were isolated as described for scRNA-seq ileal cells above, with the exception that further enrichment of viable lymphocytes by Percoll density gradient centrifugation and magnetic dead cell removal were not performed.

## Droplet-based scRNA-seq

Single-cell suspensions from the ileum and PBMCs were prepared for scRNA-seq by partitioning and preparing libraries for a target of 10,000 cells per sample according to the manufacturer's protocol for Chromium Single Cell 3′ v2 Chemistry (10X Genomics CG00052). Samples were multiplexed, had equal proportions of cDNA from each sample pooled, and were run across multiple lanes of an Illumina HiSeq 3000 with 2 × 100 paired-end sequencing as previously described (Herrera-Uribe et al, 2021). Raw data were deposited in .fastq file format for both forward and reverse strands following image analysis, base calling, and demultiplexing.

## scRNA-seq analysis of ileum data

### Initial data processing
Initial processing of scRNA-seq data included read alignment/gene quantification, ambient RNA removal, gene/cell filtering, doublet removal, normalization, integration, and dimensionality reduction identical to as previously described (Herrera-Uribe et al, 2021) and as briefly outlined below.

Read alignment, mapping, and gene quantification were carried out with the shell data package, CellRanger v4.0.0 (10X Genomics), and the *Sus scrofa* 11.1 reference genome with 11.1.97 annotation file obtained from Ensembl (Cunningham et al, 2019) and modified as previously described (Herrera-Uribe et al, 2021). Ambient RNA removal was performed using the auto-estimation method from the R package, SoupX v1.4.5 (Young & Behjati, 2020).

Genes without any reads detected in the total of all sequencing reads across all cells and samples were removed from the dataset. Cells with >12.5% of total reads attributed to mitochondrial genes, <550 total genes detected, or <1,250 total unique molecular identifiers (UMIs) detected were removed from the dataset (Table S1).

The Python package Scrublet v0.1 (Wolock et al, 2019) was used to remove highly probable neotypic doublets from the remaining dataset using a doublet rate of 0.07. Cells with corresponding doublet probability scores >0.25 were removed from our dataset (Table S1).

Normalization, integration, and dimensionality reduction were performed using the R package, Seurat v3.2.2 (Butler et al, 2018; Stuart et al, 2019). SCT-normalized data were used to perform anchor-based sample integration with default parameters. Principle component analysis (PCA) was performed to identify the first 100 principle components (PCs) of the data, and a "significant" number of PCs to use for further analyses was determined as the smaller value of (1) the highest PC that had >0.1% change in variation between consecutive PCs or (2) the smallest PC that represented >90% of the cumulative variation and <5% of variation associated with a single PC. t-distributed stochastic neighbor embedding (t-SNE) coordinates were generated for visualization using the significant number of PCs. Log-normalized and scaled data were also calculated for the RNA assay of the Seurat object.

Throughout the workflow, intermediate modified count matrices were generated and converted back to 10X format using the function *write10XCounts()* from the R package, DropletUtils v1.8.0 (Griffiths et al, 2018; Lun et al, 2019).

### Quality check of ileal sample types
Disparities in the percentage of cells removed between different ileal sample types were noted in Table S1, with smaller percentages of cells passing all cell filtering steps from non-PP (47.24% and 54.21%) compared with PP (75.31% and 77.13%) and whole ileum (71.56% and 77.40%) samples. Further investigation was performed to ensure disparities were because of differences in sample cell compositions and not differences in the quality of the same cell types between samples. Our query revealed many poor quality cells (>12.5% mitochondrial reads, <550 genes, <1,250 UMIs) that were more abundant in non-PP samples expressed genes characteristic of epithelial cells (*EPCAM*, *KRT8*) rather than immune cells (*PTPRC* [encoding pan-leukocyte marker CD45]; Fig S2A). Results were further validated by assessing the ratio of leukocytes (CD45$^+$) to epithelial cells (EPCAM$^+$) by flow cytometry (see flow cytometry methods) in independently collected ileal samples (Fig S2B and C). Non-PP samples had smaller leukocyte:epithelial cell ratios than did PP or whole-ileum samples (Fig S2D), indicating a higher occurrence of epithelial cells in non-PP samples. Similar observations were made via IHC, showing that a larger proportion of epithelial cells (stained by pan-cytokeratin protein expression; see IHC methods) were present in regions of the ileum lacking Peyer's patches (Fig S2E). Thus, smaller percentages of cells passing all cell filtering steps from non-PP samples were largely attributed to a higher occurrence of poor quality epithelial cells originally present in these samples.

### Cell clustering
The Seurat function *FindNeighbors()* was used to construct a shared nearest neighbor (SNN) graph, specifying to use the significant number of PCs calculated during data dimensionality reduction outlined above. The Seurat function *FindClusters()* using the significant number of PCs was used to identify clusters with clustering

resolutions at 0.5 intervals between 0 and 5. Clustering at different resolutions was compared using the function *clustree()* from the R package clustree v0.4.3 (Zappia & Oshlack, 2018). A clustering resolution of 3.0 was selected for all downstream work.

### Hierarchical clustering

Hierarchical clustering was performed with the Seurat function *BuildClusterTree()*, specifying to use only the number of significant PCs calculated during data dimensionality reduction outlined above for a respective dataset. Clustering dendrograms were visualized using the function *PlotClusterTree()*.

### Generating data subsets

Some analyses were conducted on only a subset of cells from the original ileum scRNA-seq dataset. To partition out only cells of interest into smaller datasets, we used the Seurat function *subset()* to specify which cells to allocate into smaller datasets. Genes with cumulative zero expression in the new data subsets, scaled data, and dimensionality reduction dimensions were removed using the function *DietSeurat()*. Data in the RNA assay were rescaled; the first 100 PCs were recalculated; the number of significant PCs was redetermined; and t-SNE dimensions were recalculated using methods described above. For some smaller data subsets, integration anchors could not be calculated with default parameter k.filter = 200 for the function *FindIntegrationAnchors()*, and the k.filter parameter was adjusted to the largest multiple of five at which integration anchors could still be calculated for the dataset.

### Cell cluster/annotation-based DGE analysis

DGE analyses were performed using functions of the Seurat package and normalized gene counts from the RNA assay. DEGs were calculated for each cluster/cell type relative to the average gene expression across an entire dataset using *FindAllMarkers()*. To be considered differentially expressed (DE), a gene was expressed in >10% of cells in one of the populations being compared, had a |logFC| >0.25, and had a corrected *P*-value <0.05.

### Multidimensional DGE analysis

Cell cluster-type-independent, multidimensional DGE analysis was performed with the R package singleCellHaystack v0.3.3 (Vandenbon & Diez, 2020). From log-normalized counts stored in the RNA assay of our Seurat object, median expression levels were calculated for each gene across the entire dataset, followed by determining if expression of each gene was above or below the median expression level within each cell. From this information, DGE was calculated using the function *haystack_highD()*, specifying to use the previously determined number of significant PCs to define the multidimensional space. A gene was considered DE if it had an adjusted *P*-value <0.05.

Gene modules were created by first selecting only DEGs with adjusted *P*-values <1 × 10$^{-10}$, followed by hierarchical clustering of selected genes with the function *hclust_haystack_highD()*. The number of gene modules to split the gene dendrogram into ($k$) was specified between $k$ = 3 to $k$ = 10 and executed with the function *cutree()*. Detection scores of each gene module within each cell were calculated with the function *plot_gene_set_haystack()*. The final value of $k$ was selected by examining each model and selecting the one producing the most interpretable and biologically relevant gene modules.

### Topic modeling

Topic models were fit with the fastTopics package v0.5-54 (Dey et al, 2017). Multiple topic models were fit for each cell type subset ($K$ = 3 to $K$ = 10). The final value of $K$ was selected by examining each model and selecting the one that produced the most interpretable and biologically relevant structure. Genes which were enriched in each topic were determined using the *diff_count_analysis()* function.

### Biological process enrichment analysis

After identification of DEGs with increased expression in cell clusters, topics, or gene modules, gene sets were subjected to Gene Ontology (GO) term enrichment analysis. GO terms were mapped to Ensembl gene IDs using the biomaRt v2.48.3 (Durinck et al, 2009) R package. Only genes, and therefore GO terms, detected in our data were included in the background set which the enriched sets were compared against. A new GO term background set was created for each data subset. Genes and GO terms which were not detected in each data subset were removed from the respective background list. The R package topGO v2.44.0 (Alexa et al, 2006) was used to carry out GO term enrichment analysis using the "elimination" algorithm and the "Fisher" test statistic. GO terms with *P*-values <0.05 were considered enriched in the gene set in question relative to the appropriate background GO term set.

### Ileal cell lineage annotations

Ileal cells were grouped into 54 clusters as described in *Cell clustering* methods (Fig S4A). Query of cell lineage canonical gene expression within clusters was used to group cells into four major cell lineages (Figs 1C and S4B and C).

### Ileal cell type annotations

Ileal cells were further annotated into 26 cell types (Figs 1D and S3) with a hybrid, multi-method approach using (1) cell clustering and accompanying cluster-based DGE and hierarchical analyses (discrete cell/non-discrete gene classifications); (2) cell cluster-independent multidimensional DGE analyses and gene module detection (non-discrete cell/discrete gene classifications); and/or (3) grade of membership/topic modeling (non-discrete cell/non-discrete gene classifications) as described below.

### T/ILC lineage lymphocyte annotations

Cells assigned as T/ILC lineage lymphocytes (Figs 1C and S4B and C) were extracted to form a subset of data for further query of T/ILC identities (as described in *Generating data subsets* methods; Fig S5A). T cell clusters were identified by expression of porcine pan-T cell marker *CD3E* (expressed by >75% of cells in a cluster; Fig S5B and C; reviewed by Piriou-Guzylack and Salmon [2008] and Gerner et al [2009]). Clusters were instead identified as ILCs if most of the cells did not express *CD3E* but still largely expressed *CD2*, similar to previously described porcine NK cells (Fig S5B and C; Herrera-Uribe et al, 2021). T cell clusters were classified as CD4 $\alpha\beta$ T cells, CD8 $\alpha\beta$ T cells, or $\gamma\delta$ T cells if >10% of cells in a cluster expressed *CD4*, *CD8B*, or *TRDC*, respectively (Fig S5D and E). Mixed CD4/CD8 $\alpha\beta$ and $\gamma\delta$/CD8 $\alpha\beta$ T cell clusters were identified if >10% of cells expressed *CD4* and *CD8B* or *CD8B* and *TRDC*, respectively (Fig S5D and E). Mixed T cell clusters appeared to be mixtures of cells expressing either *CD4*, *CD8B*, or *TRDC* rather than cells co-expressing any two of these markers (Fig S5B).

**Naive αβ T cell annotation** Further cell cluster-based DGE analysis of all T/ILC lineage lymphocytes revealed a unique gene expression signature characteristic of circulating and/or naive T cells in the only mixed CD4/CD8 αβ T cell cluster, cluster 24 (Fig S5F). Thus, cluster 24 cells were given the designation of naive CD4/CD8 αβ T cells.

**CD4 αβ T cell annotations** Cells identified as CD4 αβ T cells (Fig S5) were extracted to create a subset of data (as described in *Generating data subsets* methods). From this data subset, previous cell cluster assignments (as described in *Cell clustering* methods), topic modeling with weighted membership to three topics (as described in *Topic modeling* methods), and multidimensional DGE analysis with detection of three gene modules (as described in *Multidimensional* DGE *analysis* methods) were assessed to identify DEGs and associated enriched biological processes by each method (Fig S6A–D and Table S2). Further identification of CD4 αβ T cell populations is shown in Fig S6E–H.

**γδ/CD8 αβ T cell annotations** Cells identified as γδ T cells, CD8 αβ T cells, or a mixture of γδ/CD8 αβ T cells (Fig S5) were extracted to create a subset of data (as described in *Generating data subsets* methods). From this data subset, previous cell cluster assignments (as described in *Cell clustering* methods), topic modeling with weighted membership to three topics (as described in *Topic modeling* methods), and multidimensional DGE analysis with detection of four gene modules (as described in *Multidimensional* DGE *analysis* methods) were assessed to identify DEGs and associated enriched biological processes by each method (Fig S7A–D and Table S3). Further identification of γδ and CD8 αβ T cell types is shown in Fig S7E–J.

**ILC annotations** Cells identified as ILCs (clusters 1, 18, 43, 44, and 51; Fig S5) were extracted to create a subset of data (as described in *Generating data subsets* methods). From this data subset, previous cell cluster assignments (as described in *Cell clustering* methods), topic modeling with weighted membership to three topics (as described in *Topic modeling* methods), and multidimensional DGE analysis with detection of three gene modules (as described in *Multidimensional* DGE *analysis* methods) were assessed to identify DEGs and associated enriched biological processes by each method (Fig S8A–D and Table S4). Further identification of ILC populations is shown in Fig S8E–H.

### B lineage lymphocyte annotations
Cells identified as B lineage lymphocytes (Figs 1C and S4B and C) were extracted to create a subset of data (as described in *Generating data subsets* methods). From this data subset, previous cell cluster assignments (as described in *Cell clustering* methods), topic modeling with weighted membership to three topics (as described in *Topic modeling* methods), and multidimensional DGE analysis with detection of four gene modules (as described in *Multidimensional* DGE *analysis* methods) were assessed to identify DEGs and associated enriched biological processes by each method (Fig S9A–D and Table S5). Further identification of B lineage lymphocyte populations is shown in Fig S9E–G.

### Non-lymphocyte annotations
A data subset was created for myeloid lineage leukocytes (clusters 42, 49, 52; Fig S4) as described in *Generating data subsets* methods.

Because non-lymphocytes were not cells targeted for sequencing, cells identified as myeloid lineage leukocytes were annotated based only on cell cluster assignments (as described in *Cell clustering* methods) and associated hierarchical clustering, DGE, and biological process enrichment analyses (Fig S10A and B and Table S6). Further identification of myeloid lineage leukocyte cell types is shown in Fig S10C and D. Non-leukocytes were similarly annotated based only on cell cluster assignments (as described in *Cell clustering* methods) and associated hierarchical clustering, DGE, and biological process enrichment analyses (Fig S10E and F and Table S7). Further identification of non-leukocyte cell types is shown in Fig S10G and H.

### Reference-based label transfer and mapping
Mapping and cell label prediction of porcine ileal cells to previously annotated cells from scRNA-seq datasets was performed using Seurat v3.9.9.9026 (Butler et al, 2018; Stuart et al, 2019; Hao et al, 2021). Each porcine ileal scRNA-seq sample (n = 6) was treated as an individual query dataset, and scRNA-seq datasets of porcine PBMCs (Herrera-Uribe et al, 2021), human ileum (Elmentaite et al, 2020), and murine ileum (Xu et al, 2019) were treated as reference datasets. Post-quality control UMI count matrices, cell barcodes, and corresponding cell annotations were obtained for each reference dataset, and only cells from animals/patients under nonpathogenic/noninflammatory conditions were used to create reference datasets. The porcine PBMC dataset included cells derived from blood of steady-state pigs ranging in age between 7 wk and 12 mo (n = 7; Herrera-Uribe et al, 2021). The human ileum dataset included cells derived from ileal biopsies of eight children between 4 and 12 yr of age that did not have inflammatory bowel disease (pediatric, non-Crohn's; treated as control samples in study). Whether human ileal biopsies contained Peyer's patches was not specified, though sizeable numbers of B cells were still reported (Elmentaite et al, 2020). The murine ileum dataset included cells from the terminal ileum, enriched for cells in Peyer's patches or lamina propria (though epithelial cells still present) of specific pathogen-free BALB/cJ mice that were ~8–10 wk old. Only mice that were not induced to mount allergic responses were used (n = 14, treated as control samples in study; Xu et al, 2019). Cells with unknown identities ("unknown" annotation in porcine PBMCs and "unresolved" annotation in the murine ileum) or low UMIs ("low UMI count" designation in the murine ileum) were further removed, leaving a total of 28,684; 11,302; and 27,159 cells from porcine PBMCs, human ileum, and murine ileum, respectively. For comparison between porcine and human or murine data, genes were filtered to contain only one-to-one human-to-pig or mouse-to-pig gene orthologs, as determined with BioMart (Smedley et al, 2015) and as previously described (Herrera-Uribe et al, 2021), within both reference and query datasets. For comparison between porcine ileal and PBMC datasets, genes were not filtered further.

Each reference dataset was next converted into a Seurat object and processed with SCT normalization, integration, scaling of the integrated assay, and PCA calculations as described in preceding methods. Annotation of cell lineage was also added to cells of each reference dataset, and assignment of each annotated cell type to a cell lineage can be seen in Fig S12.

Each query dataset was processed using porcine genes for comparison to porcine PBMCs, only one-to-one human-to-pig gene

orthologs for comparison to the human ileum (termed humanized query data), or only one-to-one mouse-to-pig gene orthologs for comparison to the murine ileum (termed murinized query data), resulting in a porcine, humanized, and murinized query dataset for each of the original six ileal scRNA-seq samples (18 query datasets total). Gene names from humanized and murinized query datasets were converted to human and murine gene names, respectively. Each query dataset was then converted into a Seurat object and processed with log normalization/scaling of the RNA assay and PCA calculations as described in preceding methods. Samples of query data were processed individually and not integrated.

Prediction and mapping scores to reference datasets were next generated for query samples using integrated assay data from reference datasets and RNA assay data from query datasets. Transfer anchors between each query with a respective reference dataset were calculated using the function *FindTransferAnchors()*, with canonical correlation analysis (CCA) reduction (recommended for cross-species comparisons), log normalization, and 30 dimensions specified. Transfer anchors were used as input for the function *TransferData()* to calculate predictions probabilities (range 0–1) for each query cell at the level of cell type and cell lineage annotations, again using CCA reduction and 30 dimensions. Mapping scores (range 0–1) were also calculated with the function *MappingScore()* using the calculated transfer anchors, reference dataset cell embeddings, and query dataset neighbors, weights matrix, and embeddings as inputs.

Query data from all six samples were merged back together, and prediction probabilities and mapping scores were summarized for each respective combination of reference identity assignment and query cluster ID. Mapping scores indicated poor (score = 0) versus good (score = 1) representation of query data in a reference dataset. Prediction probabilities indicated most highly probable similar cells in the reference dataset (0 = not highly probable, 1 = highly probable).

### Pseudobulk analysis

Pseudobulk RNA-seq datasets were generated from ileum scRNA-seq samples using the Seurat function *AverageExpression()* to store mean-normalized counts from the RNA assay of Seurat objects. Pseudobulk samples were generated for each of the six ileal scRNA-seq samples either using reads after ambient RNA removal but before gene/cell filtering or using reads from the final dataset of 31,983 cells after gene/cell filtering. Using the R package, edgeR v3.30.3 (Robinson et al, 2010; McCarthy et al, 2012), pseudobulk counts were incorporated into a DGEList object with function *DGEList()*, and multidimensional scaling (MDS) plots to visualize sample-specific pseudobulk profiles were created with the function, *plotMDS()*.

### Differential abundance (DA) analysis

DA analysis of cells originating from PP versus non-PP samples of the ileum was performed using the R package, miloR v1.0.0 (Dann et al, 2021). All cells derived from PP and non-PP samples were taken as a data subset and reprocessed as described in preceding methods. In addition, SNNs were recalculated for the data subset with the Seurat function *FindNeighbors()*. The Seurat object was converted into a Milo object with incorporation of the SNN data graph calculated in Seurat with nearest neighbors (k) = 20 specified. Cell

neighborhoods were created with the miloR function *makeNhoods()*, using 20% of cells in the dataset, k = 20, and the significant number of PCs calculated for the data subset. Cell neighborhoods were visualized by overlay onto original t-SNE coordinates calculated for the original total dataset (including whole ileum samples) to promote cohesiveness with previous data visualizations. Within each cell neighborhood, the proportions of cells coming from each PP or non-PP sample were calculated. PP and non-PP samples were specified as treatment variables to compare in an experimental matrix, each with two replicates (pig 1 and 2). Distances between cell neighborhoods were calculated with *calcNhoodDistance()* and the significant number of PCs for the data subset. DA testing was then performed within each cell neighborhood with the function *testNhoods()*. Cell neighborhoods were further annotated to correspond to the 26 cell types shown in Fig 1D, where a neighborhood was assigned as a cell type if >70% of cells in the neighborhood belonged to a single respective cell type annotation. If <70% of cells in a neighborhood were annotated as a single cell type, the neighborhood was classified as mixed. A neighborhood was considered differentially abundant at a significance level of 0.1 for corrected *P*-values.

### scRNA-seq analysis of PBMC data

PBMC scRNA-seq data were processed independently but in parallel to ileal samples, with analyses mirroring those described for ileal scRNA-seq data analysis. A clustering resolution of 1.5 was used to cluster PBMCs, and cluster numbers were given a "p" prefix (e.g., cluster p1 instead of cluster 1) to differentiate from ileal cell clusters. From PBMC clusters, gene expression was assessed to identify clusters of ILCs, using the same criteria as described for identification of porcine ileal ILCs and with additional query of NK cell genes described previously (Herrera-Uribe et al, 2021). By these criteria, we identified ILCs in PBMCs as cells belonging to clusters p0, p4, p26, p28, and p30 (Fig S27).

The two PBMC samples derived from the same pigs used for ileal scRNA-seq have been published previously in the reference dataset used for porcine PBMC mapping and prediction comparisons described in above methods; however, reads were not corrected and trimmed further as was done in the previously published work (Herrera-Uribe et al, 2021).

### scRNA-seq analysis of merged ileum/PBMC ILC data

Total datasets of the porcine ileum and PBMCs were reduced using *DietSeurat()* to retain only the RNA assay and to remove dimensionality reductions, graphs, and scaled data. The two datasets were then combined with the Seurat function *merge()*, and clusters identified as ILCs from each dataset (clusters 1, 18, 43, 44, and 53 in the ileum and clusters p0, p4, p26, p28, and p30 in PBMCs) were specified within the function *subset()* to generate a data subset of only ILCs as described in *Generating data subsets* for ileal scRNA-seq analysis methods. In addition, SNNs were recalculated for the data subset with the Seurat function *FindNeighbors()*.

DA, multidimensional DGE, and biological process enrichment analyses were performed as described for ileum scRNA-seq data. DA analysis was performed between ILCs originating from ileum versus PBMC samples. Further annotation of cell neighborhoods

into specific cell types (e.g., ileal cytotoxic group 1 ILCs, peripheral ILCs) was not performed.

Gene signatures for peripheral ILCs, ileal ILCs, ileal group 1 ILCs, and group 3 ILCs were created by further filtering of genes in modules created through multidimensional DGE analysis (Table S10) that had high detection scores in ILC subsets specified in Fig 6D and E. To be included in a final gene signature, a gene from a specified module had to be expressed by >10% of cells in all target ILC subsets, and for each target ILC subset compared pairwise to each nontarget ILC subset have either (1) at least 2× the percentage of cells in the target ILC subset expressing the gene compared with the nontarget ILC subset or (2) have a larger percentage of cells (but less than 2× as many) expressing the gene and have at least 2× the average log-normalized gene expression count for the target ILC subset compared with the nontarget ILC subset. Summary statistics used for this filtering process are found in Table S10.

From this rationale, genes in gene module 4 were further filtered to identify a peripheral ILC gene signature, where peripheral ILCs were the target subset and ileal cytotoxic group 1 ILCs, ileal cycling group 1 ILCs, ileal non-naive group 1 ILCs, and ileal group 3 ILCs were nontarget subsets. Gene module 5 genes were filtered to identify an ileal ILC gene signature, where ileal cytotoxic group 1 ILCs, ileal cycling group 1 ILCs, ileal non-naive group 1 ILCs, and ileal group 3 ILCs were the target ILC subsets, and peripheral ILCs were the nontarget ILC subset. Genes in gene module 7 were filtered to identify an ileal group 1ILC gene signature, where ileal cytotoxic group 1 ILCs, ileal cycling group 1 ILCs, and ileal non-naive group 1 ILCs were target ILC subsets, and peripheral ILCs and group 3 ILCs were nontarget subsets. The ileal group 3 ILC gene signature was created by filtering genes in gene modules 3 and 8, using ileal group 3 ILCs as the target subset and ileal cytotoxic group 1 ILCs, ileal cycling group 1 ILCs, ileal non-naive group 1 ILCs, and peripheral ILCs as nontarget ILC subsets.

### Gene name modifications

Several porcine genes did not have a gene symbol assigned by Ensembl in the genome annotation file used but did have a gene symbol in the NCBI database corresponding to the Ensembl gene identifier. In these cases, we substituted the Ensembl gene identifier with the NCBI gene symbol in our figures and text and have indicated this change with an asterix (*) at the end of the gene symbol used. Genes with duplicated gene symbols were also converted to Ensembl IDs when creating our gene annotation file, and these were converted back to gene symbols in our figures and text and denoted by an asterix (*) as well. The gene annotated as *HLA-DRA* was also converted to the updated pig-specific gene symbol *SLA-DRA*. A comprehensive list of all such substitutions used in figures and text are in Table S11.

### Flow cytometry

Flow cytometry staining with cell viability dye and antibodies reactive to extracellular epitopes was performed as previously described (Wiarda et al, 2020). For antibody panels with intracellular CD79α detection, intracellular staining was completed after staining for extracellular markers, using the True-Nuclear Transcription Factor Buffer Set (424401; BioLegend) according to manufacturer's instructions and as previously described (Boettcher et al, 2020a, 2020b). Cell events were acquired on a FACSymphony A5 flow cytometer (BD Biosciences), and data were analyzed with FlowJo v10.6.1 software (FlowJo, LLC) as previously described (Wiarda et al, 2020). Fluorescence-minus-one (FMO) stains were used as controls for applying gating strategies (Fig S29).

A panel to identify general populations of leukocytes, epithelial cells, T cells, and B cells (gating strategy shown in Figs S2B and S23A) included Fixable Viability Dye-eFluor780 (65-0865-14; Thermo Fisher Scientific); mouse α-pig CD45 (MCA1222GA; Bio-Rad) detected with rat α-mouse IgG1-BUV395 (740234; BD); rat α-mouse CD326-BV605 (CD326 also known as EPCAM; 118227; BioLegend); mouse α-pig CD3ε-PE-Cy7 (561477; BD); and mouse α-human CD79α-BV421 (566225; BD). A panel to identify γδ, CD4 αβ, and CD8 αβ T cells (gating strategy shown in Fig S23B) included Fixable Viability Dye-eFluor780 (65-0865-14; Thermo Fisher Scientific); mouse α-pig CD3ε-PE-Cy7 (561477; BD); mouse α-pig γδTCR-iFluor594 (primary antibody Washington State University PG2032; custom conjugation to iFluor594 performed by Caprico Biotechnologies); mouse α-pig CD4-PerCP-Cy5.5 (561474; BD); and mouse α-pig CD8β-PE (MCA5954PE; Bio-Rad). A panel to identify ileal ILCs (gating strategy shown in Fig 5A) included Fixable Viability Dye-eFluor780 (65-0865-14; Thermo Fisher Scientific); mouse α-pig CD45 (MCA1222GA; Bio-Rad) detected with rat α-mouse IgG1-BUV395 (740234; BD); mouse α-pig CD2 (PG2009; WSU) detected with rat α-mouse IgG3-BV650 (744136; BD); mouse α-pig CD3ε-PE-Cy7 (561477; BD); mouse α-human CD79α-BV421 (566225; BD); and mouse α-pig CD172α-FITC (MCA2312F; Bio-Rad).

### Chromogenic immunohistochemistry

IHC staining for the CD3ε protein was completed on formalin-fixed, paraffin-embedded (FFPE) tissues fixed for 24–30 h in 10% neutral-buffered formalin (NBF; 3.7% formaldehyde) as previously described (Wiarda et al, 2020) with polyclonal rabbit α-human CD3ε antibody (A0452; Dako) and polyclonal goat α-rabbit HRP-conjugated antibody (K4003; Dako). The same protocol was used for CD79α protein staining but replacing the primary antibody incubation with diluted monoclonal mouse α-human CD79α antibody (LS-B4504; LifeSpan BioSciences) for 1 h RT and replacing the secondary antibody incubation with the HRP-labeled polyclonal goat α-mouse antibody (K4000; Dako) for 30 min RT. IHC staining for the pan-cytokeratin protein was carried out similar to CD3ε and CD79α IHC, but antigen retrieval was performed by incubation in Tris–EDTA buffer (10 mM Tris, 1 mM EDTA, pH 9.0; made in-house) for 20 min at 95°C. Primary antibody incubation was performed by incubating slides with diluted monoclonal mouse α-human pan-cytokeratin antibody (MCA1907T; Bio-Rad) overnight at 4°C followed by secondary antibody incubation with HRP-labeled α-mouse antibody for 30 min RT.

### Chromogenic RNA in situ hybridization

FFPE tissues were obtained as described in IHC methods. Single-color chromogenic RNA ISH with *Sus scrofa*-specific channel 1 *TRDC* probe (553141; Advanced Cell Diagnostics [ACD]) was completed with

the RNAscope 2.5 HD Reagent Kit-RED (322350; ACD) as previously described (Palmer et al, 2019) with the following modifications: (1) protease was applied for only 15 min at 40°C, and (2) slides were mounted and coverslipped as described elsewhere (Boettcher et al, 2020a). Dual chromogenic RNA ISH labeling with *Sus scrofa*-specific channel 1 (*CD8B*; ACD 815781) and channel 2 (*CD4*; 491891-C2; ACD) probes was completed with the RNAscope 2.5 HD Duplex Reagent Kit (322430; ACD) as previously described (Boettcher et al, 2020a).

### Dual protein immunofluorescence/fluorescent RNA in situ hybridization

FFPE tissues were obtained as described in IHC methods. Dual immunofluorescence (IF) labeling of CD3ε protein and fluorescent RNA ISH of *ITGAE* or *IL22* transcripts was completed according to the RNAscope Multiplex Fluorescent v2 Assay combined with Immunofluorescence-Integrated Co-Detection Workflow (ACD technical note MK 51-150), with some modifications. The RNAscope Multiplex Fluorescent Reagent Kit v2 (323100; ACD) was primarily used for RNA detection, with additional reagents outside of this kit pointed out in subsequent methods descriptions. Slides were baked, deparaffinized, and rehydrated, and hydrogen peroxide incubations were completed as described in RNA ISH methods. Target retrieval was completed by submerging slides in 1× Co-Detection Target Retrieval solution (323165; ACD) for 15 min 95°C, followed by rinsing with distilled water, 0.1% Tween in PBS (PBST), pH 7.2, and drawing of a hydrophobic barrier. CD3ε antibody (as used in IHC) was diluted in Co-detection Antibody Diluent (323160; ACD) and applied to slides for 1 h at RT, followed by washing 2 × 2 min in PBST. Tissues were fixed in 10% NBF (3.7% formaldehyde) for 30 min at RT, then washed 3 × 2 min in PBST. Protease Plus was applied to tissue sections for 15 min at 40°C, followed by rinsing with distilled water. The following steps were next completed sequentially at 40°C, with 2 × 2 min washes in 1× wash buffer between incubations: RNAscope custom channel 1 probe (*ITGAE* [590581; ACD] or *IL22* [590611; ACD]) 2 h, AMP1 30 min, AMP2 30 min, AMP3 15 min, HRP-C1 15 min, Opal 570 (FP1488001KT; Akoya Biosciences) diluted in RNAscope Multiplex TSA Buffer (ACD 322809) 30 min, HRP blocker 15 min, goat α-rabbit IgG (H+L) F(ab')-Alexa Fluor 488 (A11070; Invitrogen) diluted in Co-detection Antibody Diluent 1 h, DAPI 30 s. After DAPI incubation, DAPI was removed, and slides were mounted with ProLong Gold antifade reagent (P36930; Invitrogen) and #1.5 thickness cover glass (152450; Thermo Fisher Scientific). Slides were dried in the dark at RT, placed in the dark at 4°C overnight, and imaged within 1 wk.

Fluorescent images were acquired with a Nikon A1R Confocal Microscope using a 60× Plan Apo oil objective with numerical aperture 1.40 via hybrid galvano/resonant scanner. Single solid lasers 405, 488, and 561 were used with standard and highly sensitive GaAsP detectors with center wavelength/bandwidth 450/50 (DAPI), 525/50 (Alexa Fluor 488), and 595/50 (Opal 570). 60× large-image acquisition was used to generate a single high magnification, wide field-of-view image by automatically stitching multiple adjacent frames from a multipoint acquisition using the motorized stage on a fully automated inverted Ti2 Nikon microscope. Images were acquired and processed using NIS-Elements software (Nikon).

## Data Availability

Final data are available for download and direct query at https://singlecell.broadinstitute.org/single_cell/study/SCP1921/intestinal-single-cell-atlas-reveals-novel-lymphocytes-in-pigs-with-similarities-to-human-cells. Sequencing data are available under GEO accession GSE196388. Scripts for data analyses are available at https://github.com/USDA-FSEPRU/scRNAseq_Porcine_Ileum_PBMC. Seurat objects of processed data used for analyses are available for download and further query/analysis as .h5seurat files at https://data.nal.usda.gov/dataset/data-porcine-intestinal-innate-lymphoid-cells-and-lymphocyte-spatial-context-revealed-through-single-cell-rna-sequencing.

## Supplementary Information

## Acknowledgements

We thank the following for their excellent contributions to this work: Dr. Kristen Byrne, Zahra Bond, and Sage Becker for technical and laboratory assistance; Dr. David Alt, Dr. Mike Baker, and the Iowa State University DNA Facility for library preparation and sequencing; Drs. Daniel Nielsen and Darrell Bayles for technologic assistance; Samuel Humphrey for flow cytometry expertise and services; Judith Stasko for histology expertise and services; Adrienne Shircliff for confocal microscopy expertise and services; Drs. Nicholas Gabler and Amy Vincent for collection of postmortem samples. Research was supported by appropriated funds from USDA-ARS CRIS 5030-31320-004-00D and an appointment to the Agricultural Research Service (ARS) Research Participation Program administered by the Oak Ridge Institute for Science and Education (ORISE) through an interagency agreement between the United States Department of Energy (DOE) and the United States Department of Agriculture (USDA). ORISE is managed by Oak Ridge Associated Universities (ORAU) under DOE contract number DE-SC0014664. All opinions expressed in this paper are the authors' and do not necessarily reflect the policies and views of USDA, ARS, DOE, or ORAU/ORISE. This research used resources provided by the SCINet project of the USDA ARS project number 0500-00093-001-00-D.

### Author Contributions

JE Wiarda: conceptualization, data curation, software, formal analysis, validation, investigation, visualization, methodology, and writing—original draft.
JM Trachsel: data curation, software, and writing—review and editing.
SK Sivasankaran: data curation, software, and writing—review and editing.
CK Tuggle: conceptualization and writing—review and editing.
CL Loving: conceptualization, resources, supervision, funding acquisition, investigation, methodology, project administration, and writing—review and editing.

### Conflict of Interest Statement

The authors declare that they have no conflict of interest.

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
