## [Reviewer comments · Life Science Alliance]

Life Science Alliance

Intestinal single-cell atlas reveals novel lymphocytes in pigs with similarities to human cells

Jayne Wiarda, Julian Trachsel, Sathesh Sivasankaran, Christopher Tuggle, and Crystal Loving

DOI: <https://doi.org/10.26508/lsa.202201442>

Corresponding author(s): Crystal Loving, Agricultural Research Service - Midwest Area

Review Timeline:

Submission Date:	2022-03-10
Editorial Decision:	2022-05-03
Revision Received:	2022-07-18
Editorial Decision:	2022-07-25
Revision Received:	2022-08-04
Accepted:	2022-08-04

Transaction Report:

May 3, 2022

Re: Life Science Alliance manuscript #LSA-2022-01442-T

Dr. Crystal L Loving
Agricultural Research Service - Midwest Area
Ames

Dear Dr. Loving,

Thank you for submitting your manuscript entitled "Intestinal single-cell atlas reveals novel lymphocytes in pigs with similarities to human cells" to Life Science Alliance. The manuscript was assessed by expert reviewers, whose comments are appended to this letter. We invite you to submit a revised manuscript addressing the Reviewer comments.

Thank you for this interesting contribution to Life Science Alliance. We are looking forward to receiving your revised manuscript.

Sincerely,

B. MANUSCRIPT ORGANIZATION AND FORMATTING:

Reviewer #1 (Comments to the Authors (Required)):

This manuscript provides a single cell atlas of pig ileal leukocytes with an emphasis on lymphocyte subsets using single-cell RNAseq. Using public datasets, pig cell types were compared to their mouse and human counterparts as well as pig PBMCs. Results include a catalog of conserved features as well as species- and tissue-specific cell states and cell types. The authors use flow cytometry, immunohistochemistry, and RNA in-situ hybridization to confirm biomarkers of cell types and establish the location of cell types in Peyer's patches and intestinal epithelium. The study provides a key resource of single-cell RNAseq datasets of different components of the intestinal ileum of the pig. Such studies are important for improving swine health and for translation purposes as pigs are increasingly used as preclinical models due to their many similarities to humans. The data quality is adequate, and their data analysis methods are appropriate.

Throughout the manuscript, the authors should make the distinction between cell types and cell states. CD4 T cells and regulatory T cells are different cell types, while activated, proliferating, and naive CD8 T cells are all still CD8 T cells.

L110-111 States that pig PP are more clearly defined than in humans and rodents. Do they also make up a larger proportion of the intestine mass? If so, how does this affect the interpretation of the interspecies comparison?

Also, the manuscript should describe whether tissues collected for the mouse and human datasets differed from the samples collected to generate the pig dataset in terms of sample location, PP content, and which tissue components were collected.

L130. Explain what is meant by the "highest-resolve".

L336 It is inaccurate to state that CD2 negative gd T cells are unique to pigs.

L342 Calling CD2 negative gd T cells naïve implies that they are an intermediate of CD2+ gd T cells. While they can be stimulated to express CD2, current evidence supports that these cells represent a separate lineage of thymus-derived gd T cells with distinct tissue localization and effector functions. The text should be changed to reflect this.

L351 The statement that CD2 negative gd T cells are less well represented in mice and humans is imprecise as humans and mice don't have an equivalent of these cells.

L647-649 Can the authors comment on whether their T cell localization patterns are consistent or different compared to what has been published for mice and humans?

L718-719 There is a good chance that ITGAE+CD3- cells are not ILC1s, but dendritic cells, which should be acknowledged.

778-780 This sentence should be rephrased for clarity.

L921 The authors should add a comment that some inter-species differences observed in ileal leukocyte composition may also be due to technical reasons, such as selective death of some subsets during the isolation procedures, sampling differences, sequencing depth etc.

L960-963 It is oversimplistic to state that differences in T and B cell subsets in mucosal sites like the ileum are primarily driven by microbial exposure and other signals at these sites. It is well established that the thymus produces specialized subsets of gd and ab T cells that home to specific tissues and which are intrinsically different from conventional T cell populations. Some of the differences in subsets may due to this phenomenon.

L1544 Stipulate ages of humans and mice in public datasets and mouse strain.

L1690-1706 Flow cytometry gating strategies should include the appropriate isotype or FMO controls.

Differences in lymphocytes between pig ileum and PBMC is not surprising (Figs 2 and 3), but can the authors comment in the discussion on the similarity between their dataset and the pig lung dataset, which is another mucosal tissue?

Use of the term "activated" to describe subclusters of lymphocytes is somewhat misleading. The authors make this classification based on surface molecule expression consistent with antigen experienced or previously stimulated lymphocytes. Operationally activated lymphocytes should be enriched for cytokine and cell cycling transcripts, which these cells are not. These subsets should be named using a more precise term.

The manuscript (Fig 2B and 3B) should take advantage of well-known markers that distinguish circulating from tissue resident lymphocytes (P-selectin and L-selectin ligands), chemokine receptors and integrins (LFA1, MAC1, VLA4).

In figures 1D, 2A, and 3A, colors selected for different clusters are too similar to be able to easily distinguish clusters from each other. The authors should consider using a combination of colors that are more distinct from each other.

Some sections of the materials and methods are overly detailed, containing information already provided in the results section. The authors could make this section more succinct without losing detail about their analysis methods.

Reviewer #2 (Comments to the Authors (Required)):

In this manuscript, the authors resolved different subsets of leucocyte population in porcine ileum by single-cell-RNA-seq. Subsequently, they compared the resolved cell subsets with human and murine ileal leukocyte subsets. Their analysis identified several porcine-specific lymphocyte subsets. Finally, the authors performed in situ analysis to determine the spatial organization of the described leukocyte subsets.

The manuscript presents a descriptive study. The experimental procedures presented in adequate detail, and the figures are well-drawn. However, the results presented here would be beneficial as a reference for the leukocyte atlas of porcine ileum.

The major drawback of the manuscript is extraordinarily long length, and it would be difficult for the readers to follow, primarily due to the descript nature of the presentation. There is a redundancy of text in the introduction, result, and discussion section. The authors can shorten the length of the manuscript i) by removing redundancy in every section; ii) by moving some of the text and results (non-key results) to the supplement section of the manuscript.

The study also lacks sufficient follow-up experimental data; however, it is not necessary to add follow-up experimental.

We thank both reviewers for contributing their valuable time for review of our manuscript. We have carefully considered each critique and made suggested improvements, as reflected in our responses below. Our responses are shown in bold, while reviewer comments are not bolded.

Outside of reviewer critiques, we have also changed the term 'dysbiosis' to 'dysfunction' in the introduction (line 55), as dysbiosis refers to a microbial imbalance, while we would rather more broadly address intestinal dysregulation that is not necessarily related to the microbiota. Amendments have also been made to follow journal formatting guidelines, including renaming of Figures and Data/Tables, citation formatting, header formatting, reordering of sections, and addition of keywords, summary blurb, and running title. All figures are now found in separate files from the manuscript and figure legends attached at the end of the manuscript.

In response to reviewer comments, we refer to the line number(s) in which edits were made and paste the modified text below each comment.

Reviewer #1 (Comments to the Authors (Required)):

This manuscript provides a single cell atlas of pig ileal leukocytes with an emphasis on lymphocyte subsets using single-cell RNAseq. Using public datasets, pig cell types were compared to their mouse and human counterparts as well as pig PBMCs. Results include a catalog of conserved features as well as species- and tissue-specific cell states and cell types. The authors use flow cytometry, immunohistochemistry, and RNA in-situ hybridization to confirm biomarkers of cell types and establish the location of cell types in Peyer's patches and intestinal epithelium. The study provides a key resource of single-cell RNAseq datasets of different components of the intestinal ileum of the pig. Such studies are important for improving swine health and for translation purposes as pigs are increasingly used as preclinical models due to their many similarities to humans. The data quality is adequate, and their data analysis methods are appropriate.

Thank you for your feedback. Please see below responses to your comments.

Throughout the manuscript, the authors should make the distinction between cell types and cell states. CD4 T cells and regulatory T cells are different cell types, while activated, proliferating, and naive CD8 T cells are all still CD8 T cells.

The reviewer brings up a great example of nomenclature that lacks clarity in the field of scRNA-seq. Discussion in the article by Clevers et al. (listed below) states 'cell type' is not clearly defined in biological research and can encompass cellular states depending on the definition used. More recent annotation tools also describe multiple

levels of hierarchy for cell type annotations, including broad classification (e.g. CD4 T cell, regulatory T cell) and more narrowed classification (e.g. naïve CD8 T cell, activated CD8 T cell) that are still all considered cell types (see Conde et al. below). An explanation of how 'cell type' was used in the manuscript was added to limit misinterpretation of the subjective term and for added clarity (lines 158-160).

Lines 158-160: Cell type annotations were based on biological interpretation of genes encoding for both phenotypic and functional markers.

Clevers et al. (2017). What is your conceptual definition of "cell type" in the context of a mature organism? *Cell Systems* 4(3):255-259.

<https://doi.org/10.1016/j.cels.2017.03.006>

Conde et al. (2022) Cross-tissue immune cell analysis reveals tissue-specific features in humans. *Science* 376:713. <https://doi.org/10.1126/science.aba5197>

L110-111 States that pig PP are more clearly defined than in humans and rodents. Do they also make up a larger proportion of the intestine mass? If so, how does this affect the interpretation of the interspecies comparison?

Cross-species comparisons assessed transcriptional profiles of cells recovered in respective datasets rather than the proportions of different cell types recovered. Thus, different proportions of some cell types related to Peyer's patch abundance differences should not affect the cross-species interpretations that compared only transcriptional profiles of cells and not respective proportions. Methods used for cross-species comparisons utilized canonical correlation analyses that recognize gene expression anchors between cells of different datasets and do not require similar proportions of transcriptionally similar cells to be recovered across different samples.

Also, the manuscript should describe whether tissues collected for the mouse and human datasets differed from the samples collected to generate the pig dataset in terms of sample location, PP content, and which tissue components were collected.

Information has been added in lines 1108-1120.

Lines 1108-1120: ...only cells from animals/patients under non-pathogenic/non-inflammatory conditions were used to create reference datasets. The porcine PBMC dataset included cells derived from blood of steady-state pigs ranging in age between 7 weeks and 12 months (n = 7; Herrera-Urbe & Wiarda et al., 2021). The human ileum dataset included cells derived from ileal biopsies of eight children between 4 and 12 years of age that did not have inflammatory bowel disease (pediatric, non-Crohn's; treated as control samples in study). Whether human ileal biopsies contained Peyer's patches was not specified, though sizeable numbers of B cells were still reported (Elmentaite & Ross et

al., 2020). The murine ileum dataset included cells from terminal ileum, enriched for cells in Peyer's patches or lamina propria (though epithelial cells still present) of specific pathogen-free BALB/cJ mice that were approximately 8 to 10 weeks old. Only mice that were not induced to mount allergic responses were used (n = 14, treated as control samples in study; Xu et al., 2019).

L130. Explain what is meant by the "highest-resolve".

The sentence has been rephrased for greater clarity. Please see lines 139-141.

Lines 139-141: Collectively, the data serve as a transcriptomic atlas of the porcine intestinal immune landscape resolved at the highest level of resolution (i.e. single-cell) to date and may be used to further decode cellular phenotype and function within the intestinal tract.

L336 It is inaccurate to state that CD2 negative gd T cells are unique to pigs.

Text now denotes uniqueness only relative to humans and mice. See lines 283-284.

Lines 283-284: CD2⁻ $\gamma\delta$ T cells (characterized as *TRDC*-expressing cells that lacked *CD2* expression; Fig 2B) are present in pigs but absent from humans and mice (Stepanova and Sinkora, 2013).

L342 Calling CD2 negative gd T cells naïve implies that they are an intermediate of CD2⁺ gd T cells. While they can be stimulated to express CD2, current evidence supports that these cells represent a separate lineage of thymus-derived gd T cells with distinct tissue localization and effector functions. The text should be changed to reflect this.

As the relationship between CD2⁺ and CD2⁻ $\gamma\delta$ T cells in pigs is not fully understood, two lines of evidence exist, supporting either separate CD2⁻ and CD2⁺ cell lineages or gain of CD2 expression upon activation. The text in the manuscript was altered to provide equal presentation to both hypotheses. Please see lines 286-297.

Lines 286-297: Several lines of work support CD2⁻ $\gamma\delta$ T cells as a cell lineage separate from CD2⁺ $\gamma\delta$ T cells (Hammer et al., 2020; Sedlak et al., 2014; Sinkora et al., 2005, 2007; Stepanova & Sinkora, 2013; Rodríguez-Gómez et al., 2019), while others have suggested CD2⁻ $\gamma\delta$ T cells are naïve cells in pigs (Käser, 2021; Stepanova & Sinkora, 2012; Talker et al., 2013). We found CD2⁻ $\gamma\delta$ T cells were distantly related from all other annotated $\gamma\delta$ T cells (all considered CD2⁺ $\gamma\delta$ T cells; Fig 2B), which could suggest CD2⁻ $\gamma\delta$ T cells are a distinct cell lineage from CD2⁺ $\gamma\delta$ T cells. Contrarily, CD2⁻ $\gamma\delta$ T cells were most closely related to naïve CD4/CD8 $\gamma\delta$ T cells in porcine ileum by hierarchical clustering (Fig 2B), which could suggest CD2⁻ $\gamma\delta$ T cells are naïve cells. Regardless of whether CD2⁻ $\gamma\delta$ T cells represent a distinct cell lineage or naïve cells, CD2⁻ $\gamma\delta$ T cells had the highest average mapping scores

to porcine PBMCs of all T/ILC types, indicating CD2⁻ $\gamma\delta$ T cells to be the porcine ileal T/ILC type most similar to cells in the porcine periphery.

L351 The statement that CD2 negative $\gamma\delta$ T cells are less well represented in mice and humans is imprecise as humans and mice don't have an equivalent of these cells.

See lines 303-304 for revisions.

Lines 303-304: Thus, CD2⁻ $\gamma\delta$ T cells can be found in both ileum and periphery of pigs but do not have close counterparts in human or murine ileum.

L647-649 Can the authors comment on whether their T cell localization patterns are consistent or different compared to what has been published for mice and humans?

Comparison of T cell localization patterns to humans/mice is included in the discussion in lines 685-703.

Lines 685-703: Cycling T/ILCs, follicular CD4 $\alpha\beta$ T cells, B cells, and ASCs were enriched in Peyer's patches, suggesting induction of germinal center antibody responses characteristic of GALT. Data support findings from human research, showing an enrichment of cycling cells, CD4 $\alpha\beta$ T cells, and B cells in Peyer's patches (Fujihashi et al., 2008; Junker et al., 2009) and thus strengthening the rationale for applicability of pigs to study Peyer's patch-associated immune induction in biomedical research. Non-cycling $\gamma\delta$ T, CD8 $\alpha\beta$ T, and group 1 ILCs were enriched in the absence of Peyer's patches and detected *in situ* primarily within the ileal epithelium, suggesting these three cell types mostly comprise the intraepithelial lymphocyte (IEL) community in porcine ileum, similar to humans (reviewed by Lutter et al., 2018; Mayassi and Jabri, 2018; Olivares-Villagómez and Van Kaer, 2018). Though porcine intraepithelial group 1 ILCs have not been studied in detail, we have conducted research investigating compositional changes of intraepithelial T lymphocytes (T-IELs) in the porcine intestine after weaning (Wiarda et al., 2020). Cross-species parallels were noted in our previous work, including a predominance of CD8⁺ $\alpha\beta$ over $\gamma\delta$ T-IELs in small intestine of pigs and humans (Jarry et al., 1990; Lundqvist et al., 1995; Olivares-Villagómez and Van Kaer, 2018; Wiarda et al., 2020). In contrast, $\gamma\delta$ T-IELs are more frequent than CD8⁺ $\alpha\beta$ T-IELs in murine small intestine (Guy-Grand et al., 1991; Hoytema van Konijnenburg et al., 2017). Thus, our previous work gives initial support of pigs for IEL-centric biomedical research, and scRNA-seq of $\gamma\delta$ T, CD8 $\alpha\beta$ T, and group 1 ILCs presented herein may help to build on this application.

L718-719 There is a good chance that ITGAE+CD3⁻ cells are not ILC1s, but dendritic cells, which should be acknowledged.

This potential is now included and further discussed in lines 562-566.

Lines 562-566: Though *ITGAE*⁺*CD3ε*⁻ intraepithelial cells could also be intraepithelial dendritic cells, we found few dendritic cells expressed *ITGAE* (Fig S25C), and *ITGAE*⁺*CD3ε*⁻ cells could also be found located in the apical epithelium that was in closest proximity to the lumen, while intraepithelial dendritic cells reside only in the basement membrane (Farache et al. 2013).

778-780 This sentence should be rephrased for clarity.

Please see lines 593-594 for our alternative phrasing.

Lines 593-594: Multidimensional differential gene expression was performed to identify differentially expressed genes independent of cell type annotations and sample origin of cells (Table S10).

L921 The authors should add a comment that some inter-species differences observed in ileal leukocyte composition may also be due to technical reasons, such as selective death of some subsets during the isolation procedures, sampling differences, sequencing depth etc.

This is now mentioned in lines 680-682.

Lines 680-682: However, we also note that some inter-species differences might be attributed to technical variables, such as differences in tissue collection, cell isolation, sequencing, or data interpretation.

L960-963 It is oversimplistic to state that differences in T and B cell subsets in mucosal sites like the ileum are primarily driven by microbial exposure and other signals at these sites. It is well established that the thymus produces specialized subsets of gd and ab T cells that home to specific tissues and which are intrinsically different from conventional T cell populations. Some of the differences in subsets may due to this phenomenon.

Additional text was added to include the potential for some intestinal lymphocyte populations being directly recruited from the thymus in lines 725-733.

Lines 725-733: Though transcriptional reprogramming of T cells and ILCs upon recruitment into the intestinal tract likely contributes to transcriptional distinctions between some T cells and ILCs found in blood and intestine, other intestinal T cells and ILCs are likely unique to the intestine. For example, thymic precursors of some intestinal resident T cells are recruited directly to the intestine where they further mature as tissue resident cells (McDonald et al. 2014; Poussier et al. 1992, 1994). Therefore, findings ultimately support two potential scenarios for transcriptional distinctions between non-naive ileal and peripheral T cells and ILCs: (1) the existence of distinct cell populations exclusive to either circulation or the intestine and (2) transcriptional reprogramming of T cells and ILCs upon recruitment into the intestinal tract.

L1544 Stipulate ages of humans and mice in public datasets and mouse strain.

Information is now provided in methods (lines 1108-1120).

Lines 1108-1120: ... only cells from animals/patients under non-pathogenic/non-inflammatory conditions were used to create reference datasets. The porcine PBMC dataset included cells derived from blood of steady-state pigs ranging in age between 7 weeks and 12 months (n = 7; Herrera-Uribe & Wiarda et al., 2021). The human ileum dataset included cells derived from ileal biopsies of eight children between 4 and 12 years of age that did not have inflammatory bowel disease (pediatric, non-Crohn's; treated as control samples in study). Whether human ileal biopsies contained Peyer's patches was not specified, though sizeable numbers of B cells were still reported (Elementaite & Ross et al., 2020). The murine ileum dataset included cells from terminal ileum, enriched for cells in Peyer's patches or lamina propria (though epithelial cells still present) of specific pathogen-free BALB/cJ mice that were approximately 8 to 10 weeks old. Only mice that were not induced to mount allergic responses were used (n = 14, treated as control samples in study; Xu et al., 2019).

L1690-1706 Flow cytometry gating strategies should include the appropriate isotype or FMO controls.

We have now included all FMO information. See Supplementary Figure 29 and lines 1259-1260

Lines 1259-1260: Fluorescence-minus-one (FMO) stains were used as controls for applying gating strategies (Fig S29).

Differences in lymphocytes between pig ileum and PBMC is not surprising (Figs 2 and 3), but can the authors comment in the discussion on the similarity between their dataset and the pig lung dataset, which is another mucosal tissue?

Comparison to the pig lung dataset was considered; however, specific lymphocyte types were not mentioned in the lung dataset, past the level of identifying T cells and B cells. Therefore, we elected not to perform the comparison, as highly-resolved lymphocyte information was not published or made available for the lung data.

Use of the term "activated" to describe subclusters of lymphocytes is somewhat misleading. The authors make this classification based on surface molecule expression consistent with antigen experienced or previously stimulated lymphocytes. Operationally activated lymphocytes should be enriched for cytokine and cell cycling transcripts, which these cells are not. These subsets should be named using a more precise term.

Much like the term 'cell type', there is some interpretation and varied methods when using the term 'activated'. The term 'activated' was used to suggest both previous and current activation, and we have changed terminology to 'non-naïve' for T cells and ILCs to indicate only that cells do not have naïve transcriptional profiles. Cells annotated as 'activated B cells' were inferred to be undergoing present activation due to expression of genes such as *AICDA*, so we did not change this annotation. We have reflected changes to activated T cell and ILC nomenclature in all text, figures, and supplementary files.

The manuscript (Fig 2B and 3B) should take advantage of well-known markers that distinguish circulating from tissue resident lymphocytes (P-selectin and L-selectin ligands), chemokine receptors and integrins (LFA1, MAC1, VLA4).

Several migratory/localization markers are included in the current manuscript, including *SELL*, *ITGAE*, *CCR9*, *CCR7*, *ITGB7*, *CD69*, *S1PR1*, *KLF2*, *CXCR4*, and *CXCL8* in Figure 2B & 3B. Expression of additional markers the reviewer mentions were queried, and we've now included expression of *ITGB1* (encoding subunit of VLA-4) in Figure 2B and mentioned in-text (line 270-273). In an attempt to allow researchers to query additional genes of interest, the data is accessible for interactive, online query, as indicated in our data availability statement (lines 1340-1342).

Lines 270-273: Moreover, *SELL*^{hi} $\gamma\delta$ T cells had significantly higher expression of genes encoding for adhesion molecules (*SELL*, *ITGB1*, *ITGB7*), and the transcriptional regulator and $\gamma\delta$ T cell fate determinant, *ID3* (Fig 2B; Table S8; Lauritsen et al., 2009).

Lines 1340-1342: Final data is available for download and direct query at https://singlecell.broadinstitute.org/single_cell/study/SCP1921/intestinal-single-cell-atlas-reveals-novel-lymphocytes-in-pigs-with-similarities-to-human-cells.

In figures 1D, 2A, and 3A, colors selected for different clusters are too similar to be able to easily distinguish clusters from each other. The authors should consider using a combination of colors that are more distinct from each other.

Different color schemes were attempted, but issues deciphering all clusters were noted regardless of the colors used. Therefore, similar shades of a color were used for similar cell types (e.g. all CD4 T cells are shades of pink), as has been done in many other works (see four references from diverse journals and fields listed below). As an additional aid visualization, overlays of each cell type individually were provided (example in Supplementary Figure 3). All cell type-specific data overlaid onto t-SNE plots is also provided in additional formats, such as bar and whisker plots (e.g. Figure 3D prediction scores are shown as bar and whisker plots in Supplementary Figures 19-21).

Kashima et al. (2022). Intensive single-cell analysis reveals immune-cell diversity among healthy individuals. *Life Science Alliance* 5(7):e202201398.

<https://doi.org/10.26508/lsa.202201398>

Madisson et al. (2019). scRNA-seq assessment of the human lung, spleen, and esophagus tissue stability after cold preservation. *Genome Biology* 21(1).

<https://doi.org/10.1186/s13059-019-1906-x>

Patel et al. (2021). Single-cell resolution landscape of equine peripheral blood mononuclear cells reveals diverse cell types including T-bet+ B cells. *BMC Biology* 19:13. <https://doi.org/10.1186/s12915-020-00947-5>

Syage et al. (2020). Single-cell RNA sequencing reveals the diversity of the immunological landscape following central nervous system infection by a murine coronavirus. *Journal of Virology* 94(24):e01295-20.

<https://doi.org/10.1128/JVI.01295-20>

Some sections of the materials and methods are overly detailed, containing information already provided in the results section. The authors could make this section more succinct without losing detail about their analysis methods.

The manuscript was revised to remove redundant information, particularly in the materials and methods section.

Reviewer #2 (Comments to the Authors (Required)):

In this manuscript, the authors resolved different subsets of leucocyte population in porcine ileum by single-cell-RNA-seq. Subsequently, they compared the resolved cell subsets with human and murine ileal leucocyte subsets. Their analysis identified several porcine-specific lymphocyte subsets. Finally, the authors performed in situ analysis to determine the spatial organization of the described leucocyte subsets.

The manuscript presents a descriptive study. The experimental procedures presented in adequate detail, and the figures are well-drawn. However, the results presented here would be beneficial as a reference for the leucocyte atlas of porcine ileum.

Thank you for the feedback. To enable greater accessibility of the data as a reference atlas, we have included additional resources for data access and interactive query, as listed in the data availability statement (lines 1340-1342).

Lines 1340-1342: Final data is available for download and direct query at https://singlecell.broadinstitute.org/single_cell/study/SCP1921/intestinal-single-cell-

atlas-reveals-novel-lymphocytes-in-pigs-with-similarities-to-human-cells.

The major drawback of the manuscript is extraordinarily long length, and it would be difficult for the readers to follow, primarily due to the descript nature of the presentation. There is a redundancy of text in the introduction, result, and discussion section. The authors can shorten the length of the manuscript i) by removing redundancy in every section; ii) by moving some of the text and results (non-key results) to the supplement section of the manuscript.

The manuscript was reduced in length in some sections, especially the materials and methods. Redundancy was removed and attempts to be more concise were used.

The study also lacks sufficient follow-up experimental data; however, it is not necessary to add follow-up experimental.

July 25, 2022

RE: Life Science Alliance Manuscript #LSA-2022-01442-TR

Dr. Crystal L Loving
Agricultural Research Service - Midwest Area
1920 Dayton Avenue
Ames, IA 50010

Dear Dr. Loving,

Thank you for submitting your revised manuscript entitled "Intestinal single-cell atlas reveals novel lymphocytes in pigs with similarities to human cells". We would be happy to publish your paper in Life Science Alliance pending final revisions necessary to meet our formatting guidelines.

-please add ORCID ID for corresponding author-you should have received instructions on how to do so

Figure Check:

-please add scale bars to microscopy images

A. FINAL FILES:

B. MANUSCRIPT ORGANIZATION AND FORMATTING:

****It is Life Science Alliance policy that if requested, original data images must be made available to the editors. Failure to provide**

original images upon request will result in unavoidable delays in publication. Please ensure that you have access to all original data images prior to final submission.**

The license to publish form must be signed before your manuscript can be sent to production. A link to the electronic license to publish form will be sent to the corresponding author only. Please take a moment to check your funder requirements.

Sincerely,

Reviewer #1 (Comments to the Authors (Required)):

The authors have addressed my concerns.

August 4, 2022

RE: Life Science Alliance Manuscript #LSA-2022-01442-TRR

Dr. Crystal L Loving
Agricultural Research Service - Midwest Area
1920 Dayton Avenue
Ames, IA 50010

Dear Dr. Loving,

Thank you for submitting your Resource entitled "Intestinal single-cell atlas reveals novel lymphocytes in pigs with similarities to human cells". It is a pleasure to let you know that your manuscript is now accepted for publication in Life Science Alliance. Congratulations on this interesting work.

DISTRIBUTION OF MATERIALS:

Again, congratulations on a very nice paper. I hope you found the review process to be constructive and are pleased with how the manuscript was handled editorially. We look forward to future exciting submissions from your lab.

Sincerely,
